# PROGRESSIVE COMPOSITIONALITY IN TEXT-TO-IMAGE GENERATIVE MODELS

**Evans Xu Han**[1]    **Linghao Jin**[2]    **Xiaofeng Liu**[1]    **Paul Pu Liang**[3]

[1]Yale University, [2]University of Southern California, [3]Massachusetts Institute of Technology
`{xu.han.xh365, xiaofeng.liu}@yale.edu`
`linghaoj@usc.edu  ppliang@mit.edu`

## ABSTRACT

Despite the impressive text-to-image (T2I) synthesis capabilities of diffusion models, they often struggle to understand compositional relationships between objects and attributes, especially in complex settings. Existing approaches through building compositional architectures or generating difficult negative captions often assume a fixed prespecified compositional structure, which limits generalization to new distributions. In this paper, we argue that curriculum training is crucial to equipping generative models with a fundamental understanding of compositionality. To achieve this, we leverage large-language models (LLMs) to automatically compose complex scenarios and harness Visual-Question Answering (VQA) checkers to automatically curate a contrastive dataset, CONPAIR, consisting of 15k pairs of high-quality contrastive images. These pairs feature minimal visual discrepancies and cover a wide range of attribute categories, especially complex and natural scenarios. To learn effectively from these error cases (i.e., hard negative images), we propose EVOGEN, a new multi-stage curriculum for contrastive learning of diffusion models. Through extensive experiments across a wide range of compositional scenarios, we showcase the effectiveness of our proposed framework on compositional T2I benchmarks. The project page with data, code, and demos can be found at `https://evansh666.github.io/EvoGen_Page/`.

## 1 INTRODUCTION

The rapid advancement of text-to-image generative models (Saharia et al., 2022; Ramesh et al., 2022) has revolutionized the field of image synthesis, driving significant progress in various applications such as image editing (Brooks et al., 2023; Zhang et al., 2024; Gu et al., 2024), video generation (Brooks et al., 2024), and medical imaging (Han et al., 2024a). Despite their remarkable capabilities, state-of-the-art models such as Stable Diffusion (Rombach et al., 2022) and DALL-E 3 (Betker et al., 2023) still face challenges with *composing multiple objects into a coherent scene* (Huang et al., 2023; Liang et al., 2024b; Majumdar et al., 2024). Common issues include *incorrect attribute binding, miscounting*, and *flawed object relationships* as shown in Figure 1. For example, when given the prompt "a red motorcycle and a yellow door", the model might incorrectly bind the colors to the objects, resulting in a yellow motorcycle.

Recent progress focuses on optimizing the attention mechanism within diffusion models to better capture the semantic information conveyed by input text prompts (Agarwal et al., 2023; Chefer et al., 2023; Pandey et al., 2023). For example, Meral et al. (2023) proposes manipulating the attention on objects and attributes as contrastive samples during the test time to the optimize model performance. While more focused, the practical application of these methods still falls short of fully addressing attribute binding and object relationships. Other works develop compositional generative models to improve compositional performance, as each constituent model captures the distributions of an independent domain (Du & Kaelbling, 2024). However, these approaches assume a fixed prespecified structure to compose models, limiting generalization to new distributions.

In this paper, we argue that curriculum training is crucial to equip diffusion models with a fundamental understanding of compositionality. Given that existing models often struggle with even basic tasks (e.g., generating three cats when prompted with *"Two cats are playing"*) (Wang et al.,

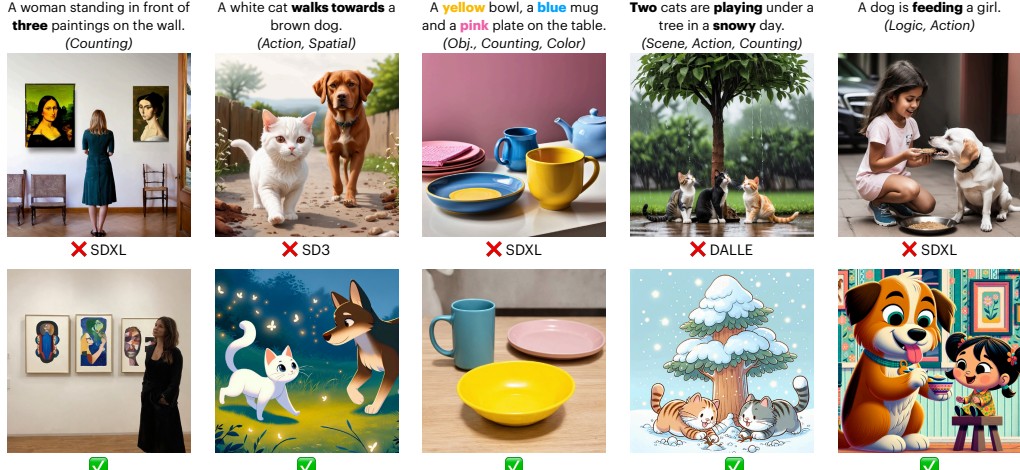

Figure 1: **Limited Compositionality Understanding in Diffusion Models**. Existing SOTA models such as SDXL, DALL-E 3 often fail to correctly compose objects and attributes. The bottom are images generated by our EVOGEN.

2024), we progressively introduce more complex compositional scenarios during fine-tuning. This staged training strategy helps models build a solid foundation before tackling intricate cases for best performance.

With the increasing demand for large-scale data in both model pre-training and fine-tuning, high-quality data generation plays a crucial role in this process (Peng et al., 2025; Ye et al., 2024; Peng et al., 2024a). Although many datasets exist for compositional generation (Wang et al., 2023; Feng et al., 2023a), there remains a significant gap in datasets that offer a clear progression from simple to complex samples within natural and reasonable contexts. Moreover, creating high-quality contrastive image datasets is both costly and labor-intensive, especially given the current limitations of generative models in handling compositional tasks. To address this, we propose an automatic pipeline to generate faithful contrastive image pairs, which we find crucial for guiding models to focus on compositional discrepancies. In summary, our work can be summarized as follows:

**Contrastive compositional dataset.** We introduce CONPAIR, a meticulously crafted compositional dataset consisting of high-quality contrastive images with minimal visual representation differences, covering a wide range of attribute categories. By leveraging LLMs, we scale up the complexity of compositional prompts while maintaining a natural context design. Our dataset features faithful images generated by diffusion models, assisted by a VQA checker to ensure accurate alignment with the text prompts.

**EVOGEN: Curriculum contrastive learning.** We also incorporate curriculum contrastive learning into a diffusion model to improve compositional understanding. This curriculum is designed with three sub-tasks: (1) learning single object-attribute composition, (2) mastering attribute binding between two objects, and (3) handling complex scenes with multiple objects. We conduct extensive experiments using the latest benchmarks and demonstrate that EVOGEN significantly boosts the model's compositional understanding, outperforming most baseline generative methods.

## 2 PRELIMINARY BACKGROUND

### 2.1 DIFFUSION MODELS

We implement our method on top of the state-of-the-art text-to-image (T2I) model, Stable Diffusion (SD) (Rombach et al., 2022). In this framework, an encoder $\mathcal{E}$ maps a given image $x \in \mathcal{X}$ into a spatial latent code $z = \mathcal{E}(x)$, while a decoder $\mathcal{D}$ reconstructs the original image, ensuring $\mathcal{D}(\mathcal{E}(x)) \approx x$.

| Dataset | # Samples | Contra. text | Contra. Image | Categories | Complex |
|---|---|---|---|---|---|
| DRAWBENCH (Saharia et al., 2022) | 200 | ✗ | ✗ | 3 (color, spatial, action) | ✓ |
| CC-500 (Feng et al., 2023a) | 500 | ✗ | ✗ | 1 (color) | ✗ |
| ATTN-AND-EXCT (Chefer et al., 2023) | 210 | ✗ | ✗ | 2 (color, animal obj.) | ✗ |
| T2I-COMPBENCH (Huang et al., 2023) | 6000 | ✗ | ✗ | 6 (color, counting, texture, shape, (non-)spatial, complex) | ✓ |
| GEN-AI (Li et al., 2024a) | 1600 | ✗ | ✗ | 8 (scene, attribute, relation, counting, comparison, differentiation, logic) | ✓ |
| ABC-6K (Feng et al., 2023a) | 6000 | ✓ | ✗ | 1 (color) | ✗ |
| WINOGROUNDT2I (Zhu et al., 2023) | 22k | ✓ | ✗ | 20 (action, spatial, direction, color, number, size, texture, shape, age, weight, manner, sentiment, procedure, speed, etc.) | ✗ |
| COMP. SPLITS (Park et al., 2021) | 31k | ✓ | ✓ | 2 (color, shape) | ✗ |
| WINOGROUND (Thrush et al., 2022) | 400 | ✓ | ✓ | 5 (object, relation, symbolic, series, pragmatics) | ✗ |
| EQBEN (Wang et al., 2023) | 250k | ✓ | ✓ | 4 (attribute, location, object, count) | ✗ |
| ARO (Yuksekgonul et al., 2023) | 50k | ✓ | ✓ | (relations, attributes) | ✗ |
| **CONPAIR (ours)** | 15k | ✓ | ✓ | 8 (color, counting, shape, texture, (non-)spatial relations, scene, complex) | ✓ |

Table 1: The comparison of compositional T2I datasets. Contra. is the abbreviation of Contrastive. *Complex* refers the samples that have multiple objects and complicated attributes and relationships.

A pre-trained denoising diffusion probabilistic model (DDPM) (Sohl-Dickstein et al., 2015; Ho et al., 2020) for noise estimation and a pre-trained CLIP text encoder (Radford et al., 2021) to process text prompts into conditioning vectors $c(y)$. The DDPM model $\epsilon(\theta)$ is trained to minimize the difference between the added noise $\epsilon$ and the model's estimate at each timestep $t$,

$$\mathcal{L} = \mathbb{E}_{z\sim\mathcal{E}(x),y,\varepsilon\sim\mathcal{N}(0,1),t} \left[ ||\varepsilon - \varepsilon_\theta(z_t, t, c(y))||_2^2 \right]. \tag{1}$$

During inference, a latent $z_T$ is sampled from $\mathcal{N}(0,1)$ and is iteratively denoised to produce a latent $z_0$. The denoised latent $z_0$ is then passed to the decoder to obtain the image $x' = \mathcal{D}(z_0)$.

## 2.2 COMPOSITIONAL DATASETS AND BENCHMARKS

The most commonly used data sets for object-attribute binding, including DRAWBENCH (Saharia et al., 2022), CC-500 (Feng et al., 2023a) and ATTEND-AND-EXCITE (Chefer et al., 2023) construct text prompts by conjunctions of objects and a few of common attributes like *color* and *shape*. To more carefully examine how generative models work on each compositional category, recent work explores the disentanglement of different aspects of text-to-image compositionality. Huang et al. (2023) introduces T2I-COMPBENCH that constructing prompts by LLMs which covers six categories including *color, shape, textual, (non-)spatial relationships* and *complex compositions*; Recently, GEN-AI (Li et al., 2024a) collects prompts from professional designers which captures more enhanced reasoning aspects such as *differentiation, logic* and *comparison*.

Another line of work proposes contrastive textual benchmarks to evaluate the compositional capability of generative models. ABC-6K (Feng et al., 2023a) contains contrast pairs by either swapping the order of objects or attributes while they focus on negative text prompts with minimal changes. WINOGROUNDT2I (Zhu et al., 2023) contains 11K complex, high-quality contrastive sentence pairs spanning 20 categories. However, such benchmarks focus on text perturbations but do not have images, which have become realistic with the advancement of generative models.

Several benchmarks featuring contrastive image pairs have also been introduced. COMPOSITIONAL SPLITS C-CUB AND C-FLOWERS (Park et al., 2021) mainly focused on the color and shape attributes of birds and flowers, sourcing from Caltech-UCSD Birds (Wah et al., 2011), Oxford-102 (Flowers) (Nilsback & Zisserman, 2008). Thrush et al. (2022) curated WINOGROUND consists of 400 high-quality contrastive text-image examples. EQBEN (Wang et al., 2023) is an early effort to use Stable Diffusion to synthesize images to evaluate the equivariance of VLMs similarity, but it lacks more complex scenarios. Yuksekgonul et al. (2023) emphasizes the importance of hard negative samples and constructs negative text prompts in ARO by swapping different linguistic elements in the captions sourced from COCO and sampling negative images by the nearest-neighbor algorithm. However, it is not guaranteed that the negative images found in the datasets truly match the semantic meaning of the prompts.

## 3 DATA CONSTRUCTION: CONPAIR

To address attribute binding and compositional generation, we propose a new high-quality contrastive dataset, CONPAIR. Next, we introduce our design principle for constructing CONPAIR.

| Category | Stage-I | Stage-II |
|----------|---------|----------|
| Shape | An **american football**. (🏈) 
 A **volleyball**. (🏐) | An american football and a volleyball. 
 A badminton ball and Frisbee. |
| Color | A blue backpack. 
 A **red** backpack | A blue backpack and a yellow purse. 
 A yellow purse and a blue backpack. |
| Counting | **Three** birds. 
 **Two** birds. | Two **cats** and one **dog**. 
 Two **dogs** and one **cat**. |
| Texture | A **plastic** toy. 
 A **fluffy** toy. | A rubber **tire** and a glass **mirror**. 
 A rubber **mirror** and a glass **tire** |
| Spatial | – | A **plate** on the right of a **bee**. 
 A **bee** on the right of a **place**. |
| Non-spatial | A basketball player is **eating dinner**. 
 A basketball player is **dancing**. | A **woman** is passing a tennis ball to a **man**. 
 A **man** is passing a tennis ball to a **woman**. |
| Scene | A **snowy** night. 
 A **rainy** night. | In a serene lake during a **thunderstorm**. 
 In a serene lake on a **sunny day**. |
| Complex | – | **Two fluffy dogs** are eating apples to the right of **a brown cat**. 
 **A brown dog** are eating pears to the left of **two fluffy cats**. |

| | Stage-III |
|---|-----------|
| Complex | Two **green** birds standing next to two **orange** birds on a **willow tree**. 
 An **orange** bird standing next to three **green** birds on the **grass**. 

 A **man** wearing a **blue** hat is throwing an **american football** from the left to the right 
 to a woman wearing a **green** pants on the playground during a **snowy day**. 
 A **woman** wearing a **green** hat is throwing a **tennis ball** from the right to the left 
 to a woman wearing a **blue** hat on the playground during a **rainy night**. |

Table 2: Examples of text prompts. Each sample has a positive (top) and a negative prompt (bottom).

Each sample in CONPAIR consists of a pair of images $(x^+, x^-)$ associated with a positive caption $t^+$.

## 3.1 GENERATING TEXT PROMPTS

Our text prompts cover eight categories of compositionality: *color, shape, texture, counting, spatial relationship, non-spatial relationship, scene*, and *complex*. To obtain prompts, we utilize the in-context learning capability of LLMs. We provide hand-crafted seed prompts as examples and predefined templates (e.g., *"A {color} {object} and a {color} {object}."*) and then ask GPT-4 to generate similar textual prompts. We include additional instructions that specify the prompt length, no repetition, etc. In total, we generate 15400 positive text prompts. More information on the text prompt generation is provided in the appendix A.

To generate a negative text prompt $t^-$, we use GPT-4 to perturb the specified attributes or relationships of the objects for Stage-I data. In Stage-II, we either swap the objects or the attributes, depending on which option makes more sense in the given context. For complex sentences, we prompt GPT-4 to construct contrastive samples by altering the attributes or relationships within the sentences. Table 2 presents our example contrastive text prompts.

## 3.2 GENERATING CONTRASTIVE IMAGES

**Minimal Visual Differences.** Our key idea is to generate contrastive images that are minimally different in visual representations. By "minimal," we mean that, aside from the altered attribute/relation, other elements in the images remain consistent or similar. In practice, we source negative image samples in two ways: 1) generate negative images by prompting negative prompts to diffusion models; 2) edit the positive image by providing instructions (e.g., change motorcycle color to red) using MagicBrush (Zhang et al., 2024), as shown at the left of Figure 2.

**Text-Image Alignment.** The high-level objective of CONPAIR is to generate positive images that faithfully adhere to the positive text guidance, while the corresponding negative images do not align with the positive text, despite having minimal visual differences from the positive images. As the quality of images generated by diffusion-based T2I generative models varies significantly (Karthik

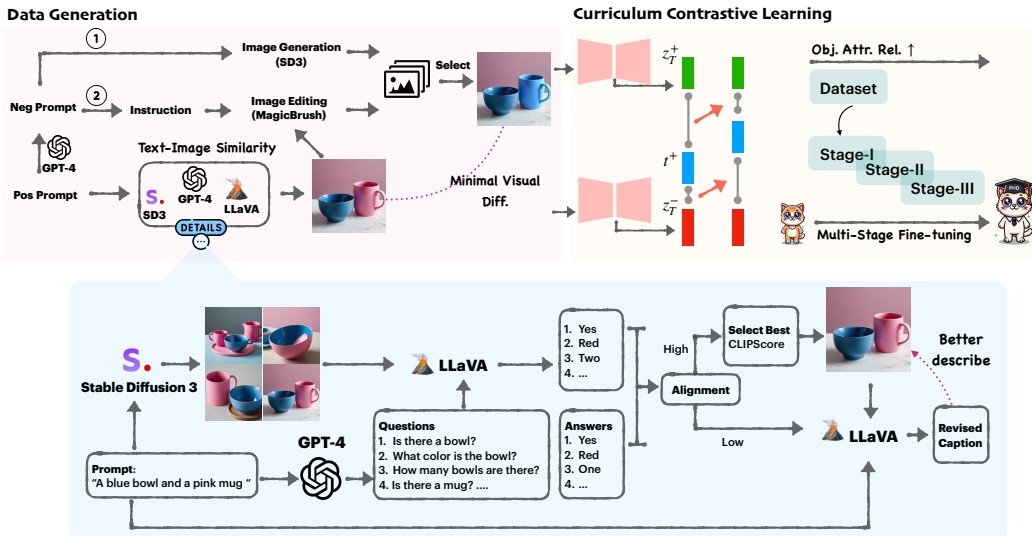

Figure 2: **EVOGEN Framework**. **Data generation pipeline** (left) **and curriculum contrastive learning** (right). **Quality control of image generation** (bottom): Given a prompt, SD3 generates multiple candidate images, which are evaluated by LLaVA. We select the best image by alignment and CLIPScore. If the alignment score is low, we prompt LLaVA to describe the image as a newly revised caption based on the generated image.

et al., 2023), we first generate 10-20 candidate images per prompt. However, selecting the most faithful image is difficult. Existing automatic metrics like CLIPScore are not always effective at comparing the faithfulness of images when they are visually similar. To address this, we propose decomposing each text prompt into a set of questions using an LLM and leverage the capabilities of VQA models to rank candidate images by their alignment score, as illustrated in Figure 2 (bottom) [1]. Note that the correct answers can be directly extracted from the prompts. Intuitively, we consider an image a success if all the answers are correct or if the alignment is greater than $\theta_{\text{align}}$ for certain categories, such as *Complex*. After getting aligned images, we select the best image by automatic metric (e.g., CLIPScore).

Empirically, we find this procedure fails to generate faithful images particularly when the prompts become *complex*, as limited by the compositionality understanding of existing generative models, which aligns with the observations of Sun et al. (2023). In response to such cases–i.e., the alignment scores for all candidate images are low–we introduce an innovative reverse-alignment strategy. Instead of simply discarding low-alignment images, we leverage a VLM to dynamically revise the text prompts based on the content of the generated images. By doing so, we generate new captions that correct the previous inaccuracies while preserving the original descriptions, thereby improving the alignment between the text and image.

**Image-Image Similarity.** Given each positive sample, we generate 20 negative images and select the one with the highest similarity to the corresponding positive image, ensuring that the changes between the positive and negative image pairs are minimal. In the case of *color* and *texture*, we use image editing rather than generation, as it delivers better performance for these attributes. Han et al. (2024b) proposes that human feedback plays a vital role in enhancing model performance. For quality assurance, 3 annotators randomly manually reviewed the pairs in the dataset and filtered 647 pairs that were obviously invalid.

## 4 EVOGEN: CURRICULUM CONTRASTIVE FINE-TUNING

A common challenge in training models with data of mixed difficulty is that it can overwhelm the model and lead to suboptimal learning (Bengio et al., 2009). Therefore, we divide the dataset into

---

[1] Examples of decomposed questions are provided in the Appendix A.3

## Attribute Binding

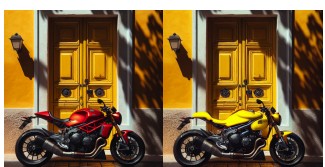
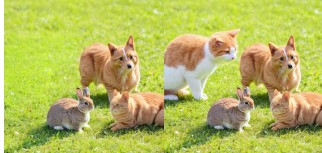
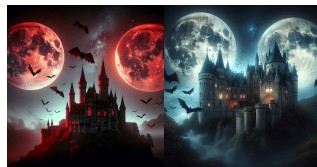

*Color*
A **red** motorcycle in front of a
**yellow** door

*Counting, Missing Object*
**Two** cats, one dog, and one rabbit
are on the grass.

*Color*
**Two blood moons** hang in the
night sky, and a flock of bats flies
over a medieval-style castle

## Object Relationships

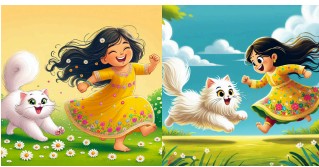
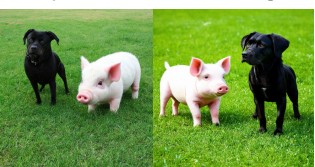
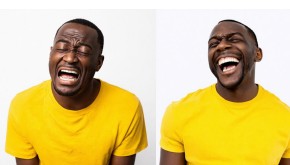

*Action*
A white **cat** is chasing a little **girl** in
a **yellow floral dress** on the grass

*Spatial*
A **black** dog is **in the left** of a pig

*Action*
A man in **yellow** T-shirt is **crying**

## Complex

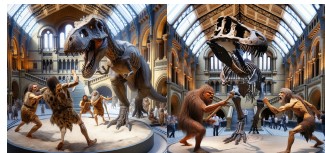
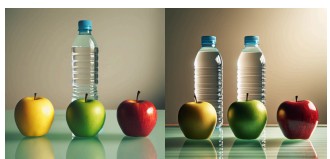
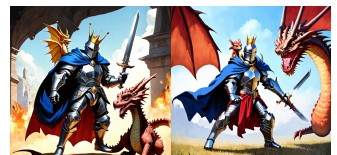

In the British Museum, **a dinosaur
fossil** is **fighting** with **four caveman**
specimens on a circular platform

**Three** differently colored **apples**
(**yellow**, **green, red** from **left to
right**) with a **transparent** water
bottle placed **behind** the **middle
apple**.

A **fully armored** knight wearing a
**blue** cape and a **small golden** dragon
perched on their **shoulder**, is **staring**
at a **red** evil dragon.

Figure 3: **Contrastive dataset** examples. Each pair includes a positive image generated from the given prompt (left) and a negative image that is semantically inconsistent with the prompt (right), differing only minimally from the positive image.

three stages and introduce a simple but effective multi-stage fine-tuning paradigm, allowing the model to gradually progress from simpler compositional tasks to more complex ones.

**Stage-I: Single object.** In the first stage, the samples consist of a single object with either a specific attribute (e.g., shape, color, quantity, or texture), a specific action, or a simple static scene. The differences between the corresponding negative and positive images are designed to be clear and noticeable. For instance, "*A man is **walking***" vs. "*A man is **eating***", where the actions differ significantly, allowing the model to easily learn to distinguish between them.

**Stage-II: Object compositions.** We compose two objects with specified interactions and spatial relationships. An example of *non-spatial relationship* is "*A **woman** chases a **dog***" vs. "*A yellow **dog** chases a **woman***." This setup helps the models learn to differentiate the relationships between two objects.

**Stage-III: Complex compositions.** To further complicate the scenarios, we propose prompts with complex compositions of attributes, objects, and scenes. Data in this stage can be: 1) contain more than two objects; 2) assign more than two attributes to each object, or 3) involve intricate relationships between objects.

Ultimately, our goal is to equip the model with the capability to inherently tackle challenges in compositional generation. Next, we discuss how to design the contrastive loss during fine-tuning at each stage. Given a positive text prompt $t$, a generated positive image $x^+$, and corresponding negative image $x^-$, the framework comprises the following three major components:

| Model | Attribute Binding | | | Object Relationship | | Complex |
|---|---|---|---|---|---|---|
| | Color | Shape | Texture | Spatial | Non-Spatial | |
| STABLE V1.4 (Rombach et al., 2022) | 37.65 | 35.76 | 41.56 | 12.46 | 30.79 | 30.80 |
| STABLE V2 (Rombach et al., 2022) | 50.65 | 42.21 | 49.22 | 13.42 | 30.96 | 33.86 |
| DALL-E 2 (Ramesh et al., 2022) | 57.00 | 55.00 | 63.74 | 13.00 | 30.00 | 37.00 |
| SDXL (Podell et al., 2023) | 64.00 | 54.00 | 36.45 | 20.00 | 31.00 | 41.00 |
| COMPOSABLE V2 (Liu et al., 2023) | 40.63 | 32.99 | 36.45 | 8.00 | 29.80 | 28.98 |
| STRUCTURED V2 (Feng et al., 2023a) | 49.90 | 42.18 | 49.00 | 13.86 | 31.11 | 33.55 |
| ATTN-EXCT V2 Chefer et al. (2023) | 64.00 | 45.17 | 59.63 | 14.55 | 31.09 | 34.01 |
| GORs (Huang et al., 2023) | 66.03 | 47.85 | 62.87 | 18.15 | 31.93 | 33.28 |
| PIXART-$\alpha$ (Chen et al., 2023) | 68.86 | 55.82 | 70.44 | 20.82 | 31.79 | 41.17 |
| MARS (He et al., 2024) | 69.13 | 54.31 | 71.23 | 19.24 | 32.10 | 40.49 |
| EVOGEN (Ours) | **71.04**$_{0.13}$ | 54.57$_{0.25}$ | **72.34**$_{0.26}$ | **21.76**$_{0.18}$ | **33.08**$_{0.35}$ | **42.52**$_{0.38}$ |

Table 3: Alignment evaluation on T2I-CompBench. We report average and standard deviations across three runs. The best results are in **bold**.

**Diffusion Model.** The autoencoder converts the positive image and negative image to latent space as $z_0^+$ and $z_0^-$. The noisy latent at timestep $t$ is represented as $z_t^+$ and $z_t^-$. The encoder of the noise estimator $\epsilon_\theta$ is used to extract feature maps $z_{et}^+$ and $z_{et}^-$ respectively.

**Projection head.** We apply a small neural network projection head $g(\cdot)$ that maps image representations to the space where contrastive loss is applied. We use a MLP with one hidden layer to obtain $h_t = g(z_{et}) = W^{(2)}\sigma(W^{(1)}(z_{et}))$.

**Contrastive loss.** For the contrastive objective, we utilize a variant of the InfoNCE loss (van den Oord et al., 2019), which is widely used in contrastive learning frameworks. This loss function is designed to maximize the similarity between the positive image and its corresponding text prompt while minimizing the similarity between the negative image and the same text prompt. The loss for a positive-negative image pair is expressed as follows:

$$\mathcal{L} = -\log \frac{\exp(\text{sim}(h_t^+, f(t))/\tau)}{\exp(\text{sim}(h_t^+, f(t))/\tau) + \exp(\text{sim}(h_t^-, f(t))/\tau)} \quad (2)$$

where $\tau$ is a temperature parameter, $f(\cdot)$ is CLIP text encoder, sim function represents cosine similarity:

$$\text{sim}(u, v) = \frac{u^T \cdot v}{\|u\|\|v\|} \quad (3)$$

This encourages the model to distinguish between positive and negative image-text pairs.

## 5 EXPERIMENTS AND DISCUSSIONS

### 5.1 IMPLEMENTATION DETAILS

**Experimental Setup** In an attempt to evaluate the faithfulness of generated images, we use GPT-4 to decompose a text prompt into a pair of questions and answers, which serve as the input of our VQA model, LLaVA v1.5 (Liu et al., 2024a). Following previous work (Huang et al., 2023; Feng et al., 2023a), we evaluate EVOGEN on Stable Diffusion v2 (Rombach et al., 2022).

**Baselines** We compare our results with several state-of-the-art methods, including trending open-sourced T2I models that trained on large training data, Stable Diffusion v1.4 and Stable Diffusion v2 (Rombach et al., 2022), DALL-E 2 (Ramesh et al., 2022) and SDXL (Podell et al., 2023). ComposableDiffusion v2 (Liu et al., 2023) is designed for conjunction and negation of concepts for pretrained diffusion models. StructureDiffusion v2 (Feng et al., 2023a), Divide-Bind (Li et al., 2024b) and Attn-Exct v2 (Chefer et al., 2023) are designed for attribute binding for pretrained diffusion models. GORs (Huang et al., 2023) finetunes Stable Diffusion v2 with selected samples and rewards. PixArt-$\alpha$ (Chen et al., 2023) incorporates cross-attention modules into the Diffusion Transformer. MARS (He et al., 2024) adapts from auto-regressive pre-trained LLMs for T2I generation tasks.

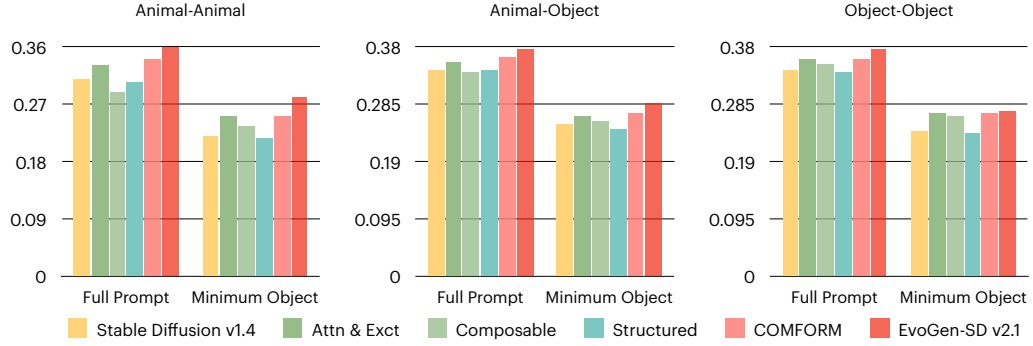

Figure 4: Average CLIP image-text similarities between the text prompts and the images generated by different models. The *Full Prompt* Similarity considers full-text prompt. *Minimum Object* represents the minimum of the similarities between the generated image and each of the two object prompts. An example of this benchmark is in subsection C.3.

**Evaluation Metrics** To quantitatively assess the efficacy of our approach, we comprehensively evaluate our method via two primary metrics: 1) compositionality on T2I-CompBench (Huang et al., 2023) [2] and 2) color-object compositionality prompts (Chefer et al., 2023).

## 5.2 PERFORMANCE COMPARISON AND ANALYSIS

**Alignment Assessment.** To examine the quality of CONPAIR, we measure the alignment of the positive image and texts using CLIP similarity. Figure 5 compares directly selecting the best image based on CLIPScore with our pipeline, which leverages a VQA model to guide image generation. These results confirm that our approach consistently improves image faithfulness across all categories with VQA assistance during image generation and demonstrate CONPAIR contains high-quality image-text pairs.

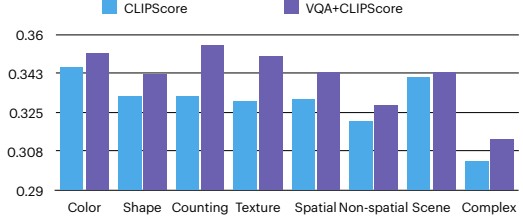

Figure 5: Average CLIP similarity of image-text pairs in CONPAIR. Applying VQA checker consistently improves text-image alignment.

**Benchmark Results** Beyond the above evaluation, we also assess the alignment between the generated images using EVOGEN and text condition on T2I-Compbench. As depicted in Table 3, we evaluate several crucial aspects, including attribute binding, object relationships, and complex compositions. EVOGEN exhibits outstanding performance across 5/6 evaluation metrics. The remarkable improvement in Complex performance is primarily attributed to Stage-III training, where high-quality contrastive samples with complicated compositional components are leveraged to achieve superior alignment capabilities.

Figure 4 presents the average image-text similarity on the benchmark proposed by Chefer et al. (2023), which evaluates the composition of objects, animals, and color attributes. Compared to other diffusion-based models, our method consistently outperforms in both *full* and *minimum* similarities across three categories, except for the minimum similarity on Object-Object prompts. These results demonstrate the effectiveness of our approach.

**Ablation Study** We conduct ablation studies on T2I-CompBench by exploring three key design choices. First, we assess the effectiveness of our constructed dataset, CONPAIR, by fine-tuning Stable Diffusion v2 directly using CONPAIR. As shown in Table 4, our results consistently outperform the baseline evaluation on Stable Diffusion v2 across all categories, demonstrating that our data generation pipeline is effective. Next, we validate the impact of our contrastive loss by comparing it

---

[2]More details about specific metrics used in T2I-CompBench are in the Appendix.

| Model | Attribute Binding | | | Object Relationship | | Complex |
|---|---|---|---|---|---|---|
| | Color | Shape | Texture | Spatial | Non-Spatial | |
| STABLE V2 (Rombach et al., 2022) | 50.65 | 42.21 | 49.22 | 13.42 | 30.96 | 33.86 |
| CONPAIR | 63.63 | 47.64 | 61.64 | 17.77 | 31.21 | 35.02 |
| CONPAIR + Contra. Loss | 69.45 | 54.39 | 67.72 | 20.21 | 32.09 | 38.14 |
| CONPAIR + Contra. Loss + Multi-stage FT | **71.04** | **54.57** | **72.34** | **21.76** | **33.08** | **42.52** |

Table 4: Ablation on T2I-CompBench. CONPAIR refers to directly finetune SDv2 on CONPAIR.

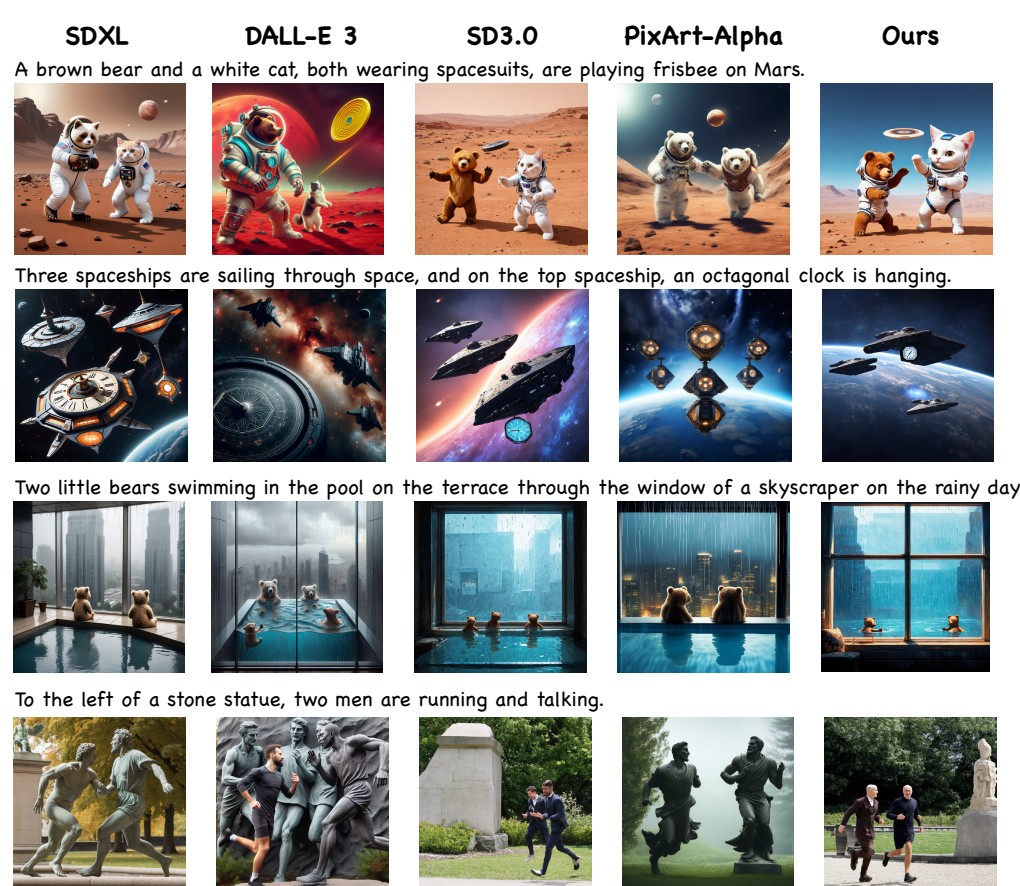

Figure 6: Qualitative comparison between EVOGEN and other SOTA T2I models. EVOGEN shows consistent capabilities in following compositional instructions to generate images.

with fine-tuning without this loss. The contrastive loss improves performance in the attribute binding category, though it has less impact on object relationships and complex scenes. We hypothesize this is because attribute discrepancies are easier for the model to detect, while relationship differences are more complex. Finally, applying the multi-stage fine-tuning strategy leads to further improvements, particularly in the *Complex* category, suggesting that building a foundational understanding of simpler cases better equips the model to handle more intricate scenarios.

**Qualitative Evaluation** Figure 6 presents a side-by-side comparison between EVOGEN and other state-of-the-art T2I models, including SDXL, DALL-E 3, SD v3 and PixArt-$\alpha$. EVOGEN consistently outperforms the other models in generating accurate images based on the given prompts. SDXL frequently generates incorrect actions and binds attributes to the wrong objects. DALL-E 3 fails to correctly count objects in two examples and misses attributes in the first case. SD v3 struggles with counting and attribute binding but performs well in generating actions. PixArt-$\alpha$ is unable to handle attributes and spatial relationships and fails to count objects accurately in the second prompt.

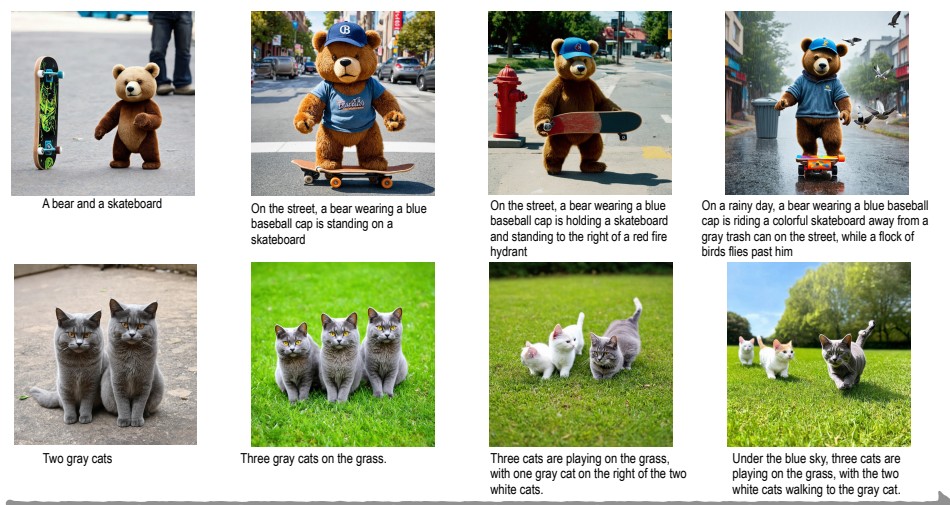

Easier, less compositionally                    More complex, more compositionally

Figure 7: Examples of EVOGEN for complex compositionality.

Figure 8: User study on 100 randomly selected prompts from Feng et al. (2023a). The ratio values indicate the percentages of participants preferring the corresponding model.

Next, we evaluate how our approach handles complex compositionality, as shown in Figure 7. Using the same object, "bear" and "cat," we gradually increase the complexity by introducing variations in attributes, counting, scene settings, interactions between objects, and spatial relationships. The generated results indicate that our model effectively mitigates the attribute binding issues present in existing models, demonstrating a significant improvement in maintaining accurate compositional relationships.

**User Study**  We conducted a user study to complement our evaluation and provide a more intuitive assessment of EVOGEN's performance. Due to the time-intensive nature of user studies involving human evaluators, we selected top-performing comparable models—DALLE-2, SD v3, SDXL, and PixArt-$\alpha$—all accessible through APIs and capable of generating images. As shown in Figure 8, the results demonstrate EVOGEN's superior performance in alignment, though the aesthetic quality may be slightly lower compared to other models.

## 6  CONCLUSION

In this work, we present EVOGEN, a curriculum contrastive framework to overcome the limitations of diffusion models in compositional text-to-image generation, such as incorrect attribute binding and object relationships. By leveraging a curated dataset of positive-negative image pairs and a multi-stage fine-tuning process, EVOGEN progressively improves model compositionality, particularly in complex scenarios. Our experiments demonstrate the effectiveness of this method, paving the way for more robust and accurate generative models.

## 7 LIMITATION

Despite the effectiveness of our current approach, there are a few limitations that can be addressed in future work. First, our dataset, while comprehensive, could be further expanded to cover an even broader range of compositional scenarios and object-attribute relationships. This would enhance the model's generalization capabilities. Additionally, although we employ a VQA-guided image generation process, there is still room for improvement in ensuring the faithfulness of the generated images to their corresponding prompts, particularly in more complex settings. Refining this process and incorporating more advanced techniques could further boost the alignment between the text and image.

## 8 REPRODUCIBILITY

We have made efforts to ensure that our method is reproducible. Appendix A provides a description of how we construct our dataset. Especially, Appndix A.1 and A.2 presents how we prompt GPT-4 and use predefined template to generate text prompts of our dataset. Appendix A.3 provides an example how we utilize VQA system to decompose a prompt into a set of questions, and answers. Appendix B provides the details of implementation, to make sure the fine-tuning is reproducible.

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

# A CONPAIR DATA CONSTRUCTION

## A.1 TEXT PROMPTS GENERATION

Here, we design the template and rules to generate text prompts by GPT-4 as follows:

- *Color*: Current state-of-the-art text-to-image models often confuse the colors of objects when there are multiple objects. Color prompts in Stage-I follow fixed sentence template *"A {color} {object}."* and *"A {color} {object} and a {color} {object}."* for Stage-II.

- *Texture*: Following Huang et al. (2023), we emphasize in the GPT-4 instructions to require valid combinations of an object and a textural attribute. The texture prompts follows the template *"A {texture} {object}."* for Stage-I and *"A {texture} {object} and a {texture} {object}."* for Stage-II.

- *Shape*: We first generate objects with common geometric shapes using fixed template *"A {shape} {object}."* for Stage-I and *"A {shape} {object} and a {shape} {object}."* for Stage-II. Moreover, we ask GPT-4 to generate objects in the same category but with different shapes, e.g., American football vs. Volleyball, as contrastive samples.

- *Counting*: Counting prompts in Stage-I follows fixed sentence template *"{count} {object}."* and *"{count} {object} and {count} {object}."* for Stage-II.

- *Spatial Relationship*: Given predefined spatial relationship such as *next to, on the left, etc*, we prompt GPT-4 to generate a sentence in a fixed template as *"{object} {spatial} {object}."* for Stage-II.

- *Non-spatial Relationship*: Non-spatial relationships usually describe the interactions between two objects. We prompt GPT-4 to generate text prompts with non-spatial relationships (e.g., actions) and arbitrary nouns. We guarantee there is only one object in the sentence for Stage-I, and two objects in Stage-II. We also find generative models fails to understand texts like *"A woman is passing a ball to a man"*. It's hard for the model to correctly generate the directions of actions. We specially design prompts like this.

- *Scene*: We ask GPT-4 to generate scenes such as weather, place and background. For Stage-I, the scene is simple, less than 5 words (e.g., on a rainy night.); For Stage-II, scenes combine weather and background or location (e.g., in a serene lake during a thunderstorm.).

- *Complex:* Here, we refer to prompts that either contain more than two objects or assign more than two attributes to each object, or involve intricate relationships between objects. We first manually curate 10 such complex prompts, each involving multiple objects bound to various attributes. These manually generated prompts serve as a context for GPT-4 to generate additional natural prompts that emphasize compositionality. The complex cases in Stage-II will be two objects with more attributes; Stage-III involves more objects.

Note that when constructing our prompts, we consciously avoided using the same ones as those in T2I-Compbench, especially considering some prompts from T2I-CompBench are empirically difficult to generate aligned image (e.g., "a pentagonal warning sign and a pyramidal bookend" as shown in Figure 9), which are not well-suited for our dataset. We have filtered out similar prompts from our dataset using LLMs to identify uncommon combinations of objects and attributes.

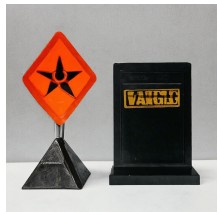

Figure 9: Example image that is hard to generate to align the prompt from T2I-CompBench.

## A.2 NEGATIVE TEXT PROMPTS GENERATION

We apply in-context learning and prompt GPT-4 to generate negative cases, we give 5-10 example test prompts each time, and make sure the generation is not repetitive, within certain lengths.

- In Stage-I, we prompt GPT-4 to change the attribute of the object such as color, shape, texture, counting, action, or scene, with instruction the differences should be noticeable.
- In Stage-II, we either swap the objects or attributes and let GPT-4 validate the swapped text prompts. For complex cases, we generate negative text by asking GPT-4 to change the attributes/relationships/scenes.
- In Stage-III, we carefully curate complicated examples with 3-6 objects, each object has 1-3 attributes, with negative prompts change attributes, actions and spatial relationships, and scenes. We also prompt GPT-4 with such examples.

## A.3 VQA ASSISTANCE

**Instruction for QA Generation**. Given an image description, generate one or two multiple-choice questions that verify if the image description is correct. Table 5 shows an example of a generated prompt and QA.

| Prompt | Question | Answer |
|---|---|---|
| A brown bear and a white cat, both wearing spacesuits, are playing frisbee on Mars | Is there a bear? | Yes |
| | Is there a cat? | Yes |
| | What color is the bear? | Brown |
| | What color is the cat? | White |
| | Does the bear wear a spacesuit? | Yes |
| | Does the cat wear a spacesuit? | Yes |
| | Is the bear playing the frisbee? | Yes |
| | Is the cat playing the frisbee? | Yes |
| | Where are they playing? | Mars |

Table 5: VQA generated questions from a prompt.

**Modifying Caption to Align Image.** Next, we illustrate how we prompt VQA to revise the caption when alignment scores of all candidate images are low. Given a generated image and an original text prompt, we prompt the VQA model with the following instruction:

Instruction: *"Given the original text prompt describing the image, identify any parts that inaccurately reflect the image. Then, generate a revised text prompt with correct descriptions, making minimal semantic changes. Focusing on counting, color, shape, texture, scene, spatial relationship, and non-spatial relationship."*. At the same time, we will provide examples of revised captions for in-context learning.

For example, given the following image (Figure 10) and the original text prompt, the modified prompt generated by the VQA model is as follows:

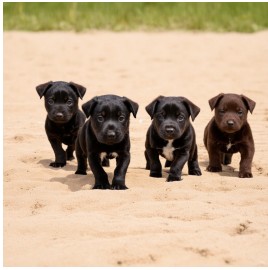

Original text prompt: **Three** puppies are **playing** on the sandy field on a sunny day, with **two** black ones **walking toward** a brown one.

Modified prompt: **Four** puppies are **standing** on a sandy field on a sunny day, with **three** black puppies and one brown puppy **facing forward**.

Figure 10: Image applies reverse-alignment.

Note that the instruction *"Focusing on the counting, color, shape, texture, scene, spatial*

*relationship, non-spatial relationship"* plays a crucial role in guiding the VQA model to provide answers that accurately correspond to the specific attributes and categories we are interested in. Without this directive, the model may occasionally fail to generate precise captions that correctly describe the image.

## A.4 DATA STATISTICS

The dataset is organized into three stages, each progressively increasing in complexity. In Stage-I, the dataset includes simpler tasks such as Shape (500 samples), Color (800), Counting (800), Texture (800), Non-spatial relationships (800), and Scene (800), totaling 4,500 samples. Stage-II introduces more complex compositions, with each category—including Shape, Color, Counting, Texture, Spatial relationships, Non-spatial relationships, and Scene—containing 1,000 samples, for a total of 7,500 samples. Stage-III repre-

|  | Stage-I | Stage-II | Stage-III | Total |
|---|---|---|---|---|
| Shape | 500 | 1000 | 200 | 1700 |
| Color | 800 | 1000 | 200 | 2000 |
| Counting | 800 | 1000 | 200 | 2000 |
| Texture | 800 | 1000 | 200 | 2000 |
| Spatial | - | 1000 | 200 | 1200 |
| Non-spatial | 800 | 1000 | 200 | 2000 |
| Scene | 800 | 1000 | 200 | 2000 |
| Complex | - | 500 | 2000 | 2500 |

Table 6: Corpus Statistics.

sents the most complex scenarios, with fewer but more intricate samples. We also include some simple cases like Stage-I and II, each contain 200 samples, while the Complex category includes 2,000 samples, totaling 3,400 samples. Across all stages, the dataset contains 15,400 samples, providing a wide range of compositional tasks for model training and evaluation. Figure 11 show more examples of images in our dataset.

## A.5 COMPARISON WITH REAL CONTRASTIVE DATASET

To evaluate how our model would fare with a real hard-negative dataset, we include the results of fine-tuning our model with COLA (Ray et al., 2023), BISON (Hu et al., 2019) evaluated by T2I-CompBench in Table 7 (randomly sampled consistent number of samples across datasets).

Although COLA and BISON try to construct semantically hard-negative queries, the majority of the retrieved image pairs are quite different in practice, often introducing a lot of noisy objects/background elements in the real images, due to the nature of retrieval from an existing dataset. We hypothesize this makes it hard for the model to focus on specific attributes/relationships in compositionality. In addition, they don't have complex prompts with multiple attributes and don't involve action, or scene.

In contrast, our dataset ensures the generated image pairs are contrastive with minimal visual changes, enforcing the model to learn subtle differences in the pair, focusing on a certain category. To the best of our knowledge, no real contrastive image dataset only differs on minimal visual characteristics.

| Dataset | Color | Shape | Texture | Spatial | Non-Spatial | Complex |
|---|---|---|---|---|---|---|
| COLA | 62.20 | 48.98 | 53.73 | 15.21 | 30.87 | 33.15 |
| BISON | 59.49 | 49.36 | 48.77 | 14.64 | 31.25 | 32.91 |
| Ours | 71.04 | 54.57 | 72.34 | 21.76 | 33.08 | 42.52 |

Table 7: Performance of fine-tuning EVOGEN on T2I-CompBench across different dataset.

## A.6 QUALITY CONTROL

**Coverage of LLM-generated QA Pairs** We conducted human evaluations on Amazon Mechanical Turk (AMT). We sampled 1500 prompt-image pairs (about 10% of the dataset, proportionally across 3 stages) to perform the following user-study experiments. Each sample is annotated by 5 human annotators.

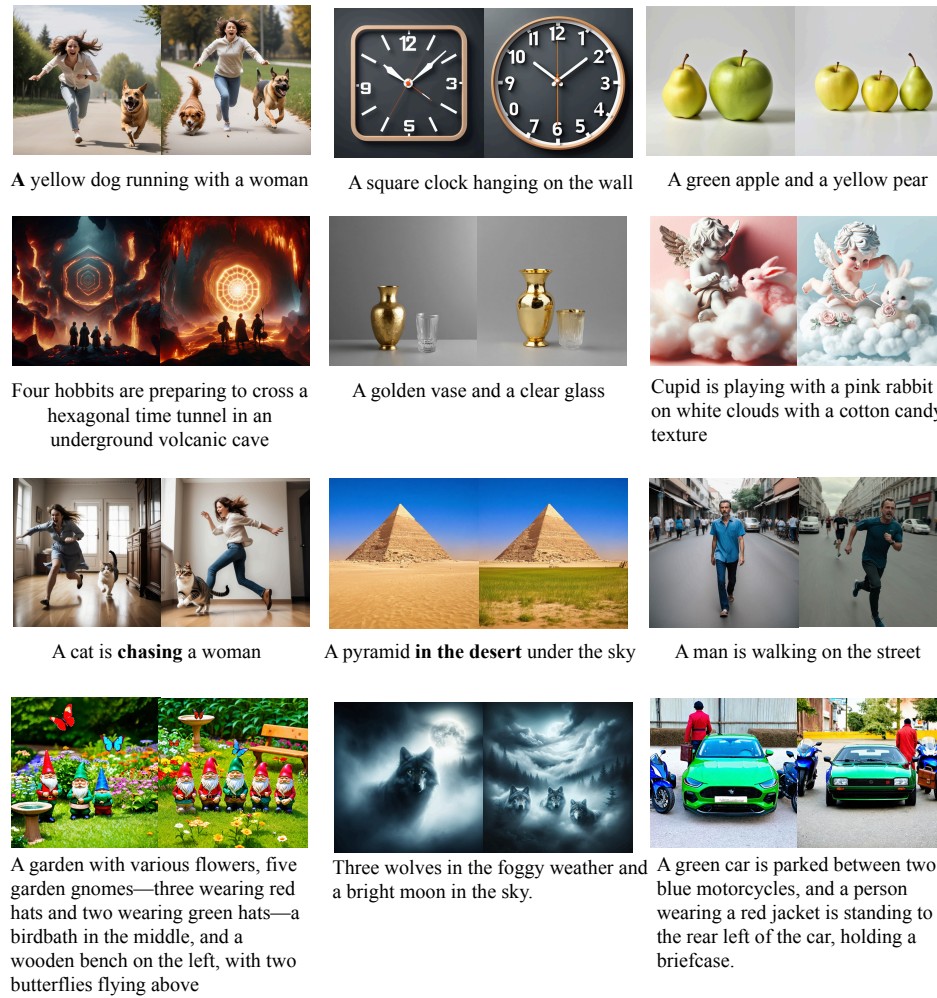

Figure 11: Example contrastive Image pairs in CONPAIR

To analyze if the generated question-answer pairs by GPT-4 cover all the elements in the prompt, we performed a user study wherein for each question-prompt pair, the human subject is asked to answer if the question-set covers all the objects in the prompt. The interface is presented in Figure 13.

Empirically, we find about 96% of the questions generated by GPT-4 cover all the objects, 94% cover all the attributes/relationships.

**Accuracy of Question-Answering of VQA Models**    To analyze the accuracy of the VQA model's answering results, we performed an additional user-study wherein for each question-image pair, the human subject is asked to answer the same question. The accuracy of the VQA model is then predicted using the human labels as ground truths. Results are displayed in Table 8.

| Image Stage | VQA Accuracy % | Annotation Time / Image (s) |
|---|---|---|
| Stage-I | 93.1% | 8.7s |
| Stage-II | 91.4% | 15.3s |
| Stage-III | 88.9% | 22.6s |

Table 8: VQA accuracy and annotation time for sampled images across different stages.

Figure 12: Comparison with Real Contrastive Dataset: COLA and BISON.

We observe that the VQA model is effective at measuring image-text alignment for the majority of questions even as the complexity of the text prompt increases, attesting the effectiveness of pipeline.

**Alignment of Revised Caption with Images**    To further validate the effectiveness of revising captions by VQA, we randomly sampled 500 images that are obtained by revising caption and performed an additional user-study for those samples that obtain low alignment score from VQA answering, but use the reverse-alignment strategy. Specifically, for each revised caption-image pair, the human subject is asked to answer how accurately the caption describes the image. The interface is presented in Figure 14. Note we have 5 annotators, each is assigned 100 caption-image pairs.

Empirically, we found that 4% of the samples show that the revised caption similarly describes the image as the original caption. 94.6% of the samples show the revised caption better describes the image. Overall,with the following settings, the average rating of the alignment between revised caption and image is 4.66, attesting that revised caption accurately describes the image.

**Prompt**

A bear and a white cat, both wearing spacesuits, are playing frisbee on Marks.

**Question Set**

Is there a bear?
Is there a cat?
What color is the bear?
What color is the cat?
Does the bear wear a spacesuit?
Does the cat wear a spacesuit?
Is the bear playing the frisbee?
Is the cat playing the frisbee?
Where are they playing?

Does the above question set covers all the objects and relationships/attributes in the prompt?

◉ All the objects and relationships/attributes are covered.

◯ All the objects are covered, some relationships/attributes are missing.

◯ Some objects are missing, all the relationships/attributes are missing.

◯ Some objects are missing, some relationships/attributes are missing.

Figure 13: Interface for User Study: Coverage of LLM-generated QA Pairs

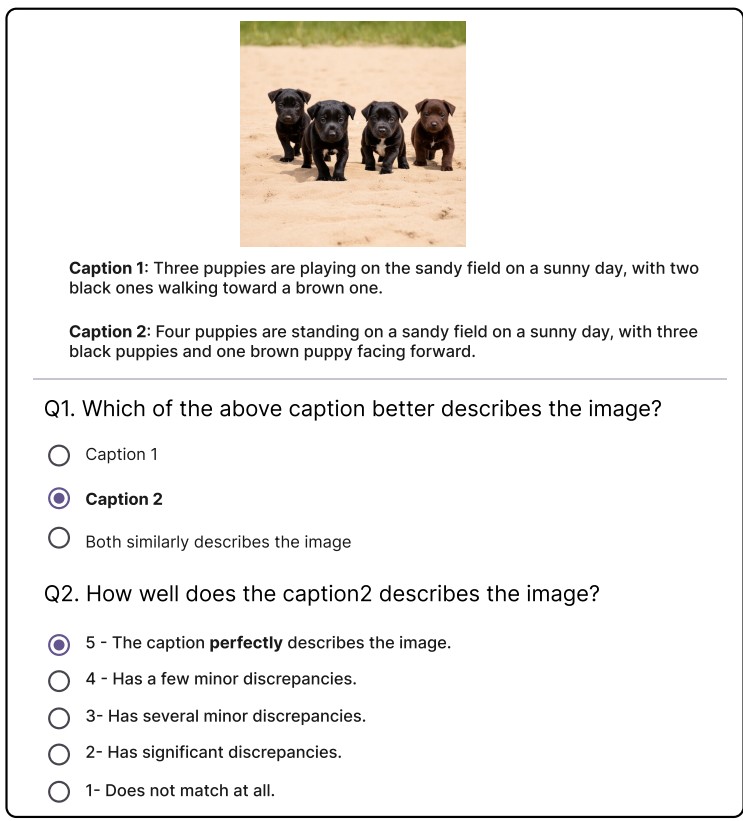

**Caption 1**: Three puppies are playing on the sandy field on a sunny day, with two black ones walking toward a brown one.

**Caption 2**: Four puppies are standing on a sandy field on a sunny day, with three black puppies and one brown puppy facing forward.

Q1. Which of the above caption better describes the image?

◯ Caption 1

◉ **Caption 2**

◯ Both similarly describes the image

Q2. How well does the caption2 describes the image?

◉ 5 - The caption **perfectly** describes the image.

◯ 4 - Has a few minor discrepancies.

◯ 3- Has several minor discrepancies.

◯ 2- Has significant discrepancies.

◯ 1- Does not match at all.

Figure 14: Interface for User Study: Alignment of Revised Caption with Images

**Similarity of Contrastive Image Pairs**    We have 3 annotators in total, each annotator is assigned 2550 images (about 50% samples) to check if the positive and negative image pairs align with its text prompt and are similar with small visual changes on specific attributes/relationships. We filtered 647 images from the randomly selected 7650 images, which is 8.45%, attesting the quality of most images in the dataset.

| Method | Basic | | | | | | Advanced | | | | | |
| --- | --- | --- | --- | --- | --- | --- | --- | --- | --- | --- | --- | --- |
| | Attribute | Scene | Relation | | | Avg | Count | Differ | Compare | Logical | | Avg |
| | | | Spatial | Action | Part | | | | | Negate | Universal | |
| SD v2.1 | 0.75 | 0.77 | 0.72 | 0.72 | 0.69 | 0.74 | 0.66 | 0.63 | 0.61 | 0.50 | 0.57 | 0.58 |
| SD-XL Turbo | 0.79 | 0.82 | 0.77 | 0.78 | 0.76 | 0.79 | 0.69 | 0.65 | 0.64 | **0.51** | 0.57 | 0.60 |
| DeepFloyd-IF | 0.82 | 0.83 | 0.80 | 0.81 | 0.80 | 0.81 | 0.69 | 0.66 | 0.65 | 0.48 | 0.57 | 0.60 |
| SD-XL | 0.82 | 0.84 | 0.80 | 0.81 | 0.81 | 0.82 | 0.71 | 0.67 | 0.64 | 0.49 | 0.57 | 0.60 |
| Midjourney v6 | 0.85 | 0.88 | 0.86 | 0.86 | 0.85 | 0.85 | 0.75 | 0.73 | 0.70 | 0.49 | 0.64 | 0.65 |
| SD3-Medium | 0.86 | 0.86 | 0.87 | 0.86 | 0.88 | 0.86 | 0.74 | 0.77 | 0.72 | 0.50 | **0.73** | 0.68 |
| DALL-E 3 | **0.91** | **0.91** | 0.89 | 0.89 | **0.91** | **0.90** | 0.78 | 0.76 | 0.70 | 0.46 | 0.65 | 0.65 |
| EvoGen- SD3-Medium (Ours) | 0.89 | 0.88 | **0.90** | **0.91** | 0.88 | 0.89 | **0.80** | **0.79** | **0.73** | **0.51** | **0.73** | **0.72** |

Table 9: Gen-AI Benchmark Results.

# B  Training Implementation Details

We implement our approach upon Stable Diffuion v2.1 and Stable Diffusion v3-medium. We employ the pre-trained text encoder from the CLIP ViT-L/14 model. The VAE encoder is frozen during training. The resolution is 768, the batch size is 16, and the learning rate is 3e-5 with linear decay.

# C  Quantitative Results

## C.1  T2I-CompBench Evaluation Metrics

Following T2I-CompBench, we use DisentangledBLIP-VQA for color, shape, texture, UniDet for spatial, CLIP for non-spatial and 3-in-1 for complex categories.

## C.2  Gen-AI Benchmark

We further evaluate EvoGen on the Gen-AI (Li et al., 2024a) benchmark. For a fair comparison with DALL-E 3, we finetune our model on Stable Diffusion v3 medium. As indicated in Table 9, EvoGen performs best on all the *Advanced* prompts, although it exhibits relatively weaker performance in some of the basic categories compared to DALL-E 3.

## C.3  Attn & Exct Benchmark Prompt Examples

The benchmark protocol we follow comprises structured prompts 'a [animalA] and a [animalB]', 'a [animal] and a [color][object]', 'a [colorA][objectA] and a [colorB][objectB]' . Table 10 demonstrate the results of average CLIP similarities between text prompts and captions generated by BLIP for Stable Diffusion-based methods on this benchmark. EvoGen outperforms those models in three categories.

| Model | Animal-Animal | Animal-Obj | Obj-Obj |
| --- | --- | --- | --- |
| Stable v1.4 (Rombach et al., 2022) | 0.76 | 0.78 | 0.77 |
| Composable v2 (Liu et al., 2023) | 0.69 | 0.77 | 0.76 |
| Structured v2 (Feng et al., 2023a) | 0.76 | 0.78 | 0.76 |
| Attn-Exct v2 (Chefer et al., 2023) | 0.80 | 0.83 | 0.81 |
| CONFORM (Meral et al., 2023) | 0.82 | 0.85 | 0.82 |
| Ours | 0.84 | 0.86 | 0.85 |

Table 10: Attn-Exct benchmark Results.

# D  Qualitative Results

Figure 15 presents more comparison between EvoGen and other state-of-the-art T2I models, including SDXL, DALL-E 3, SD v3 and PixArt-$\alpha$.

# E  Related Work

With the rapid development of multimodal learning (Li et al., 2023; Liang et al., 2024b;a; Han et al., 2025) and image generation (Yu et al., 2024; Liu et al., 2024b; Weber et al., 2024; Peng et al., 2024b; Kim et al., 2025), understanding and addressing compositional challenges in text-to-image generative models has been a growing focus in the field (Thrush et al., 2022; Huang et al., 2023; Chefer et al., 2023; Peng et al., 2024c). In particular, Zarei et al. (2024) identifies key compositional

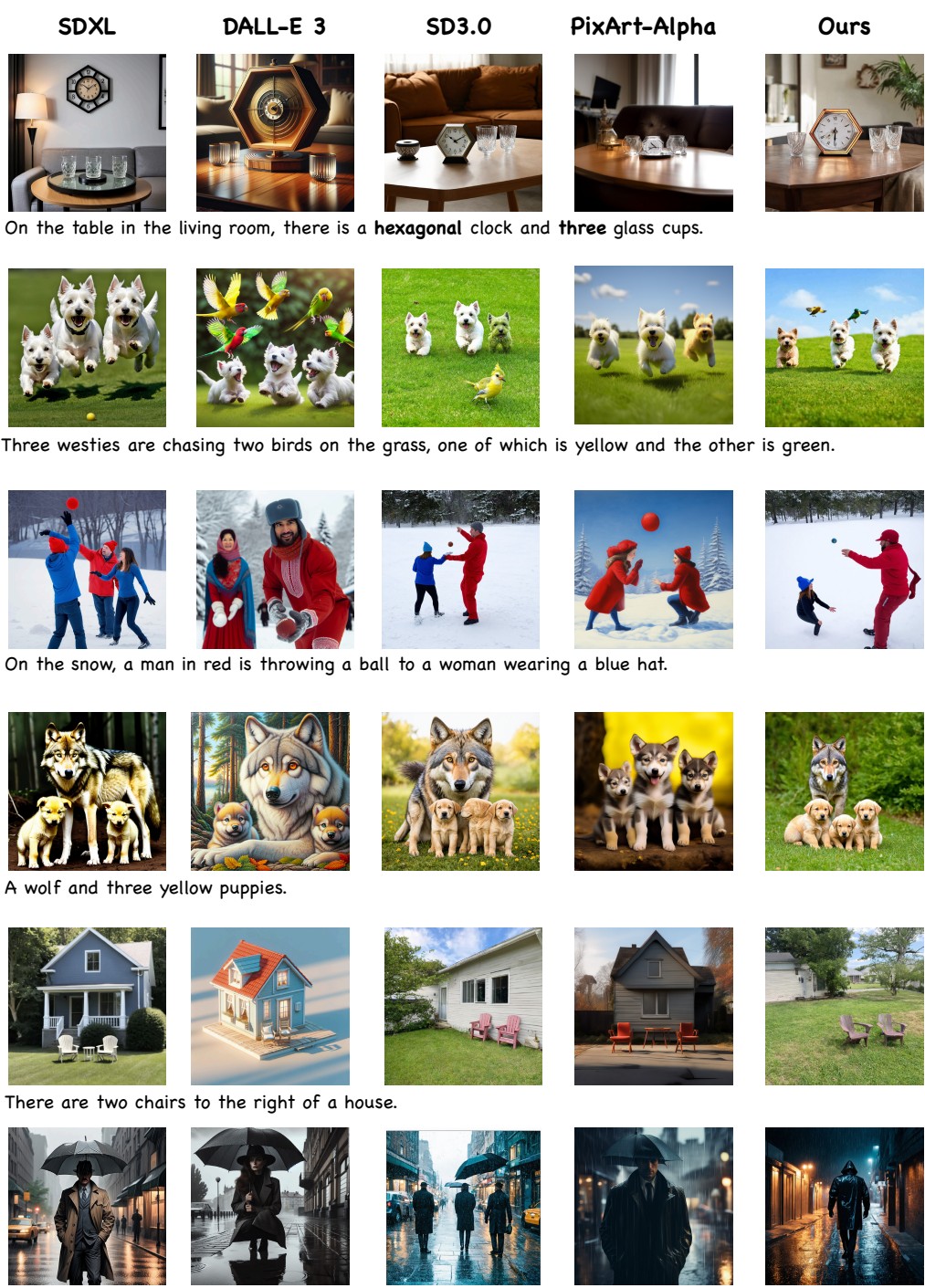

Figure 15: Qualitative Results.

challenges in text-to-image diffusion models and proposes strategies to enhance attribute binding and object relationships. Leveraging the power of large-language models (LLMs) for compositional generation is another area of active research (Drozdov et al., 2022; Mitra et al., 2024; Pasewark et al., 2024). For instance, Feng et al. (2023b) leverages large language models (LLMs) to generate visually coherent layouts and improve compositional reasoning in visual generation tasks.

