# OpenReview forum: "Progressive Compositionality in Text-to-Image Generative Models"
_ICLR.cc/2025/Conference — ICLR 2025 Spotlight_

### Official Review · Reviewer_Amfr · 2024-11-02

**Soundness:** 3
**Presentation:** 3
**Contribution:** 3
**Rating:** 6
**Confidence:** 4

**Summary:**

This paper introduces CONTRAFUSION, a novel framework designed to improve the compositional understanding of text-to-image (T2I) diffusion models. The authors address these models' common issues, such as incorrect attribute binding and flawed object relationships, by introducing COM-DIFF, a high-quality contrastive dataset consisting of minimally differing image pairs across diverse attribute categories. Using a curriculum-based contrastive learning approach, CONTRAFUSION progressively fine-tunes models from simple to complex compositional scenarios. By leveraging large-language models (LLMs) and Visual-Question Answering (VQA) systems for enhanced image-text alignment, the framework significantly boosts model performance on compositional benchmarks, setting a new standard for T2I generation.

**Strengths:**

The paper introduces a novel methodology targeting the major shortcomings of image generation models, such as the lack of expressive capability for multiple objects or attributes. It explains the creation of a benchmark to address these issues and the application of contrastive learning for model fine-tuning to achieve improvement, demonstrating superior performance across various models.

**Weaknesses:**

1. The argument that three different sub-tasks were resolved solely using contrastive learning is not very convincing, especially since the paper does not propose a novel loss function but rather uses an existing one. Given that the main contribution of this work is the creation of a benchmark, more justification is needed to support the claim of achieving multiple improvements solely from this contribution.

2. With the text prompts structured into eight categories, there seems to be potential for generating a higher number of combinations than the 15,000 currently provided, which feels somewhat limited.

3. The method used to measure text-image alignment has already been employed in several papers, such as TIFA and other works evaluating text-to-image generation using VQA and LLMs. These approaches have limitations, such as not being able to account for all elements in the generated images using a limited QA set, and issues like hallucination from LLMs and VLMs. It appears that these concerns have not been adequately addressed.

4. The idea of modifying text prompts to match generated images, as mentioned around line 250, is unclear. It’s difficult to imagine how such prompts would be presented, especially given that generated images often have subtle and complex characteristics that are hard to describe accurately, as exemplified by the top-right part of Figure 2.

5. The description of the three stages appears for the first time on page 5, but sufficient explanation should be provided earlier, such as in Table 2 or preceding sections.

6. The claim that three annotators randomly assessed image-image similarity is questionable, as it would have been better to review all images comprehensively. There are likely cases where the differences were not properly captured. Alternatively, some automated method should have been used to ensure accuracy.

**Questions:**

1. How do you ensure that when generating negative images, only the specified attribute or relationship described in the prompt was changed? It’s unclear how this was rigorously verified.

2. In Figure 3, isn't the dog's head in the second image of the first row incorrectly generated? It looks like there might be an issue with the image.

3. What is the training cost of the proposed approach? Considering the computational expense, it’s not entirely clear if the performance improvement is significant enough to justify the cost.

4. Are there any adverse effects, such as a decrease in the model's ability to generate diverse images, after fine-tuning?

5. Given that CLIPScore has several limitations as an evaluation metric, such as being inaccurate in counting objects, isn't it contradictory to use CLIPScore for measuring alignment in a paper that aims to overcome these limitations?

---

> ### Author Response · Authors · 2024-11-20
> **Response to Reviewer Amfr (Part 1)**
>
> We thank Reviewer Amfr for their time and feedback! We are encouraged that R4 finds that “our methodology  is novel” and “our framework improves compositional performance”. We appreciate the insightful questions, which have helped clarify certain aspects of our work. Below, we will address R4’s feedback point-by-point.
>
> >**W1: More justification is needed to support the claim of achieving multiple improvements solely from this contribution.**
>
> Thanks for the feedback! According to the ablation study in Table 4, fine-tuning COM-DIFF without our CONTRAFUSION framework (which combines the designed contrastive loss with a multi-stage fine-tuning strategy) yields only limited improvements. Introducing the contrastive loss allows the model to effectively utilize contrastive image pairs, while the multi-stage fine-tuning strategy further enhances performance, especially in the *complex* category.
>
> | Model Setup| Color | Shape | Texture | Spatial | Non-Spatial | Complex |
> | ------- | ------ | ------ | ------- | ------- | ------- | ------- |
> | COM-DIFF | 63.63 |  47.64 |  61.64  | 17.77  | 31.21 | 35.02 |
> | COM-DIFF + Contra. Loss | 69.45 | 54.39 | 67.72 | 20.21 | 32.09 | 38.14 |
> | COM-DIFF + Contra. Loss + Multi-stage FT | 71.04 | 54.57 | 72.34 | 21.76 | 33.08 | 42.52 |
>
> As it is critical to keep our arguments precise, we will clarify the wording in our revision.
>
> >**W2: there seems to be potential for generating a higher number of combinations than the 15,000 currently provided..**
>
> At 15,000 contrastive image pairs, our Com-Diff dataset is indeed smaller than some other benchmark datasets. However, our aim is to keep high-quality contrastive image-pairs with minimal changes, which limits dataset size due to resource and costs.
>
> A high-level objective of our work is to introduce an automatic pipeline to generate contrastive image pairs and demonstrates that fine-tuning with such image pairs can serve as a realistic and useful setting to improve compositionality of T2I models. We are definitely open to expanding the dataset and improving our dataset quality, and perform scaling law on how data size will impact our model performance in future work.
>
> >**W3: Limitations from LLM and VLMs**
>
> We recognize existing work leveraging earlier LLMs and VQA models may suffer from this problem. However, given a text prompt consisting of a fixed number of objects and attributes and is fixed in pattern in most cases, the GPT-4 generated questions-answer pairs in our method empirically cover basically all/most the elements by decomposing the prompt, indicating current SOTA LLMs is able to handle such tasks more effectively.
>
> To ensure the quality of our dataset, we conduct fine-grained analysis and rigorous experiments below (more details regarding annotation process and interface can be found  in Appendix A.6).
>
> - **User study of element coverage in the LLM-generated QA pairs.**
>
> We sampled 1500 prompts (about 10% of our dataset,  proportionally across 3 stages) and performed additional user study by asking annotators if the GPT-4 generated question set covers all the elements in the prompt. Each sample is annotated by 5 human annotators. The interface is presented in Figure 13.
>
> Empirically, we find about 96% of the questions generated by GPT-4 cover all the objects, 94% cover all the attributes/relationships.
>
> - **User study of accuracy of question-answering from VQA models.**
>
> As noted in the response to W1@Reviewer PM7r, we performed an additional user study wherein for each question-image pair, the human subject is asked to answer the same question. The accuracy of the VQA model is then predicted using the human labels as ground truths.
>
> Results are shown below (more details about data quality evaluation are in Appendix A.6), we observe that the VQA model is able to effectively measure alignment even when the text prompt becomes complex.
>
>
> | Image Stage | VQA Accuracy % | Annotation Time / Image (s) |
> | ----- |---------------- | -------------------- |
> | Stage-I | 93.1% | 8.7s |
> | Stage-II | 91.4% | 15.3s |
> | Stage-III | 88.9% | 22.6s |
>
> Overall, we also see significant improvements in compositionally ability evaluated by T2I-CompBench (Table 3) obtained in our current framework by using LLMs and VQA models.
>
> We appreciate R3's suggestion on evaluating the quality of our dataset, we've added more details and interfaces about human evaluations in Appendix A.6.

---

> ### Author Response · Authors · 2024-11-20
> **Response to Reviewer Amfr (Part 2)**
>
> >**W4: The idea of modifying text prompts to match generated images, line 250, is unclear.**
>
> Thanks for the feedback! Though it might intuitively look difficult to accurately describe complex prompts by revising captions with VQA models, we empirically show how this procedure works with examples. More details about the procedure can be found in Appendix A.3.
>
> We use the following instruction to revise captions:
>
> **Instruction**: Given the original text prompt describing the image, identify any parts that inaccurately reflect the image. Then,generate a revised text prompt with correct descriptions, making minimal semantic changes. Focusing on the counting, color, shape, texture, scene, spatial relationship, non-spatial relationship. (also give examples of revised prompts).
>
> Given an image (Figure 10) and  a text prompt, *“**Three** puppies are **playing** on the sandy field on a sunny day, with two black ones **walking toward** a brown one.”*
>
> The prompt is modified to *“**Four** puppies are **standing** on a sandy field on a sunny day, with **three** black puppies and one brown puppy **facing forward**. ”*
>
> The revised prompt made minimal semantic changes to the original prompt, but correctly described the images. Even such subtle characteristics like not “ playing on”  but “standing” can be detected by the VQA model, attesting the effectiveness of our proposed method. We also found that without specific instruction to ask VQA to focus on the categories we are interested in, although VQA is powerful,  the model may occasionally fail to generate precise captions that correctly describe the image.
>
> To rigorously validate our claim and evaluate how accurately the revised caption describes the image, we randomly sampled 500 prompt-image pairs, that are selected by revising the caption, and performed additional user study by asking annotators how accurately the caption describes the image (interface shown in Figure 14). Note we have 5 annotators, each is assigned 100 caption-image pairs.
>
> Empirically, we found that 4% of the samples show that the revised caption similarly describes the image as the original caption. 94.6% of the samples show the revised caption better describes the image. Overall,with  the following settings, the average rating of the alignment between revised caption and image is 4.66, attesting that revised caption accurately describes the image.
>
> - 5: The caption perfectly describes the image.
> - 4: Has a few minor discrepancies.
> - 3: Has several minor discrepancies.
> - 2: Has significant discrepancies.
> - 1: Does not match at all.
>
> We appreciate the valuable feedback and have added the above discussion in our revision (Appendix A.3 and A.6).
>
> >**W5: The description of the three stages appears for the first time on page 5, but…**
>
> Thank you for your valuable suggestion. Though we briefly mention curriculum learning in the last paragraph of introduction, we agree that we should introduce multi-stage training in more detail earlier. We’ll adjust the order in our revision!
>
>
> >**W6: Image Similarity Quality Control**
>
> Details about the image similarity quality control are answered to W1@Reviewer PM7r. We manually examined 50% of the dataset, and filtered out approximately 8% image pairs that is either not well-aligned with text prompts or not similar. Empirically, the dataset contains over 95% high-quality contrastive pairs. We are definitely open to improving the quality of our dataset by comprehensively examining it in the future!
>
> - Automatic Metrics:
>
> We also evaluate the image similarity with cosine similarity between the image embeddings (CLIP Image Similarity). The average similarity score across 15,000 image pairs is 85.7%, indicating that the image pairs are highly similar, with only small differences.
>
> >**Q1: How do you ensure that when generating negative images, only the specified attribute was changed?**
>
> Given the positive prompt, we edit it to the negative prompt with only specified attribute and relationship changes/swap, which ensure minimal changes semantically. We also employed image editing to change specific attributes in the image to keep minimal visual changes.
>
> To validate the similarity between positive and negative image pairs, we have human evaluators to filter those image pairs that are not similar. We also show that the CLIP image  similarity between image pairs is 86.7%,  rigorously attesting the quality of the rest of the dataset.

---

> ### Author Response · Authors · 2024-11-20
> **Response to Reviewer Amfr (Part 3)**
>
> >**Q2: In Figure 3, isn't the dog's head in the second image of the first row incorrectly generated?**
>
> We appreciate your careful evaluation. We will discuss more about this in the limitation section.
>
> Such artifacts are a common limitation of all current generative models, including SOTA  models. However, addressing this issue is not one of the primary objectives of our work. To maintain the quality of our dataset, we implemented rigorous filtering procedures, including both automated checks and manual reviews. However, given the dataset’s scale, a small number of edge cases (<5%, based on a recent audit) may have slipped through. These cases represent only a minor fraction, and the vast majority of examples in the dataset exhibit high fidelity in their compositional representation.
>
> >**Q3: What is the training cost of the proposed approach? if the performance improvement is significant enough to justify the cost**
>
> Our fine-tuning procedure is computationally efficient with 15,000 samples. However, with our pipeline and constructed dataset, the improvement on most evaluation benchmarks are significant, which is worthwhile compared to the costs. We believe our dataset could also benefit future research and the proposed pipeline is also valuable to improve T2I compositionality.
>
>
> >**Q4: Are there any adverse effects, such as a decrease in the model's ability to generate diverse images, after fine-tuning**
>
> Our paper primarily focuses on addressing compositionality in text-to-image generative models, and as such, diversity is not the main focus of our work. We believe that enhancing compositionality does not necessarily diminish the model’s ability to generate diverse images, as our method targets alignment without constraining the variety of generated outputs.
>
> Empirically, we calculate FID on 10000 images randomly sampled from the COCO 2014 validation set, which might give some insights about the diversity of generated images. The results are shown below.
>
> | Model | FID |
> | ------ | ---- |
> | SD v2.1 | 13.89 |
> | Ours | 11.14 |
>
> We observe that the image quality and diversity remain stable, as measured by the FID score of the generated images.
>
> >**Q5: Given that CLIPScore has several limitations as … isn't it contradictory to use CLIPScore for measuring alignment in a paper that aims to overcome these limitations?**
>
> In our evaluation (Table 3), T2I-Compbench incorporates multiple metrics beyond  CLIPScore, including BLIP-VQA, Uni-Det and 3-in-1, as noted in the response to Q2@Reviewer cqxf, to complements the limitations of CLIPScore.
>
> Although the limitations of CLIPScore, it’s widely used in existing works and serves as a useful metric for demonstrating improvements in alignment with our method (Figure 4). We also present results with human annotations in Appendix A.6, to provide a more comprehensive evaluation.
>
>  We sincerely appreciate R4 for your valuable reviews and suggestions, we have added those elaborations in our revision.

---

> ### Author Response · Authors · 2024-11-24
> **Follow-up to Rebuttal**
>
> Dear Reviewer Amfr,
>
> Thank you again for your valuable feedback and acknowledgement of our work. We'd like to check in with you and see if our rebuttal has clarified your concerns. If there are still any unclear parts in our work, please let us know and we would be happy to engage further before the discussion period closes. Thanks for your time and effort in reviewing this paper!

---

> > ### Comment · Reviewer_Amfr · 2024-11-26
> >
> > Thank you for your sincere response. Regarding Q2, the intention was not to address such limitations but to suggest replacing the image with a better one. As for Q3, the question is about which GPU was used and how many hours the training took.

---

> > > ### Author Response · Authors · 2024-11-26
> > > **Response to Reviewer Amfr**
> > >
> > > Dear Reviewer Amfr,
> > >
> > > Thank you for reading our rebuttal and for your thoughtful response.
> > >
> > > For Q2, we have updated Figure 3 in the revised PDF with a higher-quality image pair from our dataset. Specifically, the image describes *Two cats, one dog, and one rabbit are on the grass*, but one of the cat is missing in the contrastive image.
> > >
> > > For Q3, the model was trained on 8 H100 GPUs, with a total training time of approximately 10 hours. We believe the training costs are reasonable given the observed improvements.
> > >
> > > We sincerely appreciate your insightful questions, as they have helped us make meaningful improvement to our paper.

---

> > > > ### Comment · Reviewer_Amfr · 2024-11-26
> > > >
> > > > Thank you for response, I'll raise the score

---

> > > > > ### Author Response · Authors · 2024-11-26
> > > > > **Response to Reviewer Amf**
> > > > >
> > > > > We genuinely appreciate the time and efforts you dedicated to reviewing our paper. We will further polish it in the final revision. Thank you!

---

### Official Review · Reviewer_cqxf · 2024-11-04

**Soundness:** 3
**Presentation:** 3
**Contribution:** 3
**Rating:** 8
**Confidence:** 4

**Summary:**

The paper introduces ContraFusion, a framework designed to enhance the compositional understanding of diffusion models through a combination of contrastive learning and multi-stage fine-tuning. This approach addresses the prevalent challenges faced by text-to-image (T2I) synthesis models, particularly in managing complex relationships between objects and attributes. It introduces the Com-Diff dataset, which comprises 15,000 carefully curated contrastive image pairs, tailored to refine the model's performance.

**Strengths:**

- The paper addresses a critical issue in text-to-image generative models, focusing on the challenge of compositionality. Despite significant advancements in this field, even the most sophisticated models often struggle with accurately rendering complex compositional details. This work provides an approach to mitigate these limitations.
- While datasets for compositionality already exist, this paper makes a valuable contribution by introducing a new dataset that features contrastive pairs of images generated from positive and negative prompts. This dataset is specifically designed to enhance model performance on compositional tasks, making it a valuable resource for further research.
- The paper applies two techniques—contrastive loss and multi-stage training—for fine-tuning Stable Diffusion on compositional data. It demonstrates the effectiveness of each technique through quantitative analysis, providing evidence of their impact on improving model accuracy and handling of compositional elements.

**Weaknesses:**

- The ContraFusion model is compared against other methods using the T2I-CompBench dataset. However, it is trained on the Com-Diff dataset, which likely overlaps noticeably with the T2I-CompBench test set. To address this, I recommend that the authors provide a detailed analysis of any overlap between Com-Diff and T2I-CompBench and discuss its potential impact on the reported results. Furthermore, I suggest evaluating the model on a completely separate test set to ensure the robustness and generalizability of the findings.
- For improved clarity, it would be beneficial if the paper explicitly stated the metrics used in Table 3 and other relevant sections. For instance, the BLIP VQA score is employed for evaluation in Table 3, but this is not directly mentioned. Clearly indicating this metric would help readers better understand the basis of the model’s evaluation and the reported results.
- The paper would benefit from more comprehensive details about the experimental setup. For example, it should specify the number of samples generated for calculating the BLIP VQA score, discuss the potential variability in results by indicating the standard deviation across different seeds, and provide a more detailed description of the user study, including the number of participants involved.
- Incorporating a dedicated subsection on relevant background research would provide a richer context for understanding the current contributions. Specifically, discussing recent methodologies like those in [1] and [2], which offer new perspectives on addressing compositionality issues, would demonstrate awareness of and engagement with the latest developments in the field.

[1] LayoutGPT: Compositional Visual Planning and Generation with Large Language Models, Feng et al, 2023.

[2] Understanding and Mitigating Compositional Issues in Text-to-Image Generative Models. Zarei et al, 2024.

**Questions:**

- Do you have evaluation results for training on the Com-Diff dataset using multi-stage fine-tuning without incorporating contrastive loss?
- Regarding the multi-stage fine-tuning process, have you considered reintroducing a small number of samples from earlier stages to maintain performance on simpler compositional prompts? Observations from Table 4 suggest a larger performance gap in complex scenarios (excluding texture); thus, recycling simpler samples might help bridge this gap and enhance overall model robustness.

---

> ### Author Response · Authors · 2024-11-20
> **Response to Reviewer cqxf (Part 1)**
>
> We thank Reviewer cqxf for their time and feedback! We are encouraged that R3 finds that our work “mitigates a critical issue in text-to-image generative models” and that “our dataset is  a valuable resource for further research”. We will respond point-by-point to R3’s feedback below:
>
> >**W1: Overlap between Com-Diff and T2I-CompBench; Evaluating the model on a completely separate test set**
>
> We agree that our work could definitely benefit from a detailed analysis of the overlap with T2I-CompBench and evaluations on other benchmarks. Below, we include those two result:
>
> - **Overlap with T2I-CompBench**
>
> When constructing our prompts, we consciously avoided using the same ones as those in T2I-Compbench, especially considering some prompts from T2I-CompBench are empirically difficult to generate aligned image (e.g., “a pentagonal warning sign and a pyramidal bookend” as shown in Figure 9), which are not well-suited for our dataset. We have filtered out similar prompts from our dataset using LLMs to identify uncommon combinations of objects and attributes.
>
> Specifically, during the prompt construction process, approximately 1% of our prompts overlapped with those from T2I-Compbench, which have also been filtered out.
>
> - **GenAI benchmark**
>
> We also evaluate our model on the Gen-AI benchmark. To ensure a fair comparison with DALL-E 3, we fine-tune our model on SD v3. As shown below, ContraFusion outperforms all other models on advanced prompts. While it demonstrates slightly weaker performance in some basic categories compared to DALL-E 3, it still surpasses other models, highlighting the effectiveness of our framework.
>
> We have added those two analysis in our revision (See more details in Appendix A.1 & Appendix C.2).
>
> |                          |               |           | **Basic**   |            |          |          |           |            | **Advanced** |            |               |          |
> |--------------------------|---------------|-----------|-------------|------------|----------|----------|-----------|------------|--------------|------------|---------------|----------|
> | **Method**               | **Attribute** | **Scene** | **Spatial** | **Action** | **Part** | **Avg**  | **Count** | **Differ** | **Compare**  | **Negate** | **Universal** | **Avg**  |
> | SD-XL Turbo              | 0.79          | 0.82      | 0.77        | 0.78       | 0.76     | 0.79     | 0.69      | 0.65       | 0.64         | **0.51**   | 0.57          | 0.60     |
> | DeepFloyd-IF             | 0.82          | 0.83      | 0.80        | 0.81       | 0.80     | 0.81     | 0.69      | 0.66       | 0.65         | 0.48       | 0.57          | 0.60     |
> | SD-XL                    | 0.82          | 0.84      | 0.80        | 0.81       | 0.81     | 0.82     | 0.71      | 0.67       | 0.64         | 0.49       | 0.57          | 0.60     |
> | Midjourney v6            | 0.85          | 0.88      | 0.86        | 0.86       | 0.85     | 0.85     | 0.75      | 0.73       | 0.70         | 0.49       | 0.64          | 0.65     |
> | SD3-Medium               | 0.86          | 0.86      | 0.87        | 0.86       | 0.88     | 0.86     | 0.74      | 0.77       | 0.72         | 0.50       | **0.73**      | 0.68     |
> | DALL-E 3                 | **0.91**      | **0.91**  | 0.89        | 0.89       | **0.91** | **0.90** | 0.78      | 0.76       | 0.70         | 0.46       | 0.65          | 0.65     |
> | Ours | 0.89          | 0.88      | **0.90**    | **0.91**   | 0.88     | 0.89     | **0.80**  | **0.79**   | **0.73**     | **0.51**   | **0.73**      | **0.72** |
>
>
> >**W2: Explicitly stated the metrics used in Table 3 and other relevant sections …**
>
> Thanks for the valuable suggestion! We agree explicitly stating the metrics and important considering the difficulty of evaluating compositionality in generative models. Following [1], we use DisentangledBLIP-VQA[1] for color, shape, texture, UniDet[2] for spatial, CLIP for non-spatial and 3-in-1 [1] for complex categories. In Figure 4 and Figure 5, we use average CLIP score between image and prompt similarity.
>
> We have clarified this point in our revision (Appendix C.1).
>
> [1] Huang, et al., T2I-CompBench: A Comprehensive Benchmark for Open-world Compositional Text-to-image Generation
> [2] Zhou, et al., Simple multi-dataset detection

---

> ### Author Response · Authors · 2024-11-20
> **Response to Reviewer cqxf (Part 2)**
>
> >**W3: More comprehensive details about experiment setup**
>
> For fair comparison, we generate 10 images for each text prompt for T2I-CompBench evaluation. We present the standard deviation results in Table 3 and will run other experiments with different seeds to get standard deviation.
>
> - **Human Evaluation Details**
>
> As answered in the response to W1@PM7r, we tasked 3 annotators in total, each annotator assigned 2550 images out of the total 7650 images (about 50% samples, proportionally across three stages), to check if the positive and negative images align with the text prompt well and are similar with minimal changes. We filtered only 647 images from the selected 7650 images, which is 8.45%, attesting the quality of the dataset.
>
> To further guarantee the quality of our dataset, we perform additional user-studies in terms of *coverage of LLM-generated QA pairs*, *accuracy of question-answering of VQA models* and *alignment of revised caption with images* with 1500 randomly selected samples with 5 annotators. More details and interfaces can be found in **Appendix A.6**.
>
> >**W4: Relevant background research**
>
> We strongly agree that adding relevant background research is important to the authors to understand this field and our contributions.  Due to the limitation of words, we didn’t add related work in the submission.
>
> Both of the references provide insightful ideas regarding mitigating compositionality issues. We thank R3 for providing these two references, and we have incorporated both references into the related work section in our revision (Appendix E).
>
> >**Q1: Do you have evaluation results for training on the Com-Diff dataset using multi-stage fine-tuning without incorporating contrastive loss?**
>
> To further disentangle the impact of multi-stage and contrastive loss, we present the results using solely multi-stage without contrastive loss below.
>
> | Model Setup| Color | Shape | Texture | Spatial | Non-Spatial | Complex |
> | ------- | ------ | ------ | ------- | ------- | ------- | ------- |
> | COM-DIFF | 63.63 |  47.64 |  61.64  | 17.77  | 31.21 | 35.02 |
> | COM-DIFF + Multi-stage FT| 67.11 | 49.95 | 62.24 | 19.81 | 31.39 | 37.43 |
> | COM-DIFF + Contra. Loss | 69.45 | 54.39 | 67.72 | 20.21 | 32.09 | 38.14 |
> | COM-DIFF + Contra. Loss + Multi-stage FT | 71.04 | 54.57 | 72.34 | 21.76 | 33.08 | 42.52 |
>
> Solely applying multi-stage doesn’t significantly improve the performance, we hypothesize this is due to the model not being able to fully explore the Com-Diff dataset without contrastive loss. We will add the results in Table 4.
>
> >**Q2: Recycling simpler compositional prompts**
>
> We appreciate R3’s valuable suggestion and find it quite inspiring to prevent the model forgetting the generative ability on simple prompts by recycling simpler samples. We plan to conduct experiments by introducing varying proportions of simple prompts in later stages and will include these results in an ablation study once we have determined the optimal settings.

---

> > ### Comment · Reviewer_cqxf · 2024-11-27
> >
> > Thank you for your detailed responses, as well as for providing additional results and elaborations. One of my primary concerns was the potential overlap between the evaluation dataset of T2I-CompBench and the training data used by the authors. Based on your response, it is reassuring to hear that there is no such overlap after filtering. For future work, however, I recommend providing a distinct test partition of the dataset to ensure greater transparency and reproducibility for subsequent studies.
> >
> > Additionally, based on my experience with T2I-CompBench, generating only 10 samples per evaluation may lead to less stable results. I would suggest increasing the number of samples in future experiments to enhance reliability, perhaps to 50 or even 100 samples.
> >
> > I also appreciate the results you provided for COM-DIFF + Multi-Stage FT (without Contrastive Loss). These results clearly demonstrate that each proposed method contributes to improving the compositional performance of the model.
> >
> > Considering all these points, I would like to raise my score to reflect the improvements and clarifications provided.
> >
> > Thank you again for your efforts.

---

> > > ### Author Response · Authors · 2024-11-28
> > > **Response to Reviewer cqxf**
> > >
> > > Thank you for your thoughtful feedback and for taking the time to provide detailed suggestions and recommendations. We greatly appreciate your kind words and recognition of the efforts we have made to address your concerns. We'll further polish our  future work based on your suggestions!

---

> ### Author Response · Authors · 2024-11-24
> **Follow-up to Rebuttal**
>
> Dear Reviewer cqxf,
>
> Thank you again for your valuable feedback and acknowledgement of our work. We'd like to check in with you and see if our rebuttal has clarified your concerns. If there are still any unclear parts in our work, please let us know and we would be happy to engage further before the discussion period closes. Thanks for your time and effort in reviewing this paper!

---

### Official Review · Reviewer_drFx · 2024-11-04

**Soundness:** 4
**Presentation:** 4
**Contribution:** 3
**Rating:** 8
**Confidence:** 4

**Summary:**

The authors make a dataset of images with hard negatives for a text prompt. They use LLMs to generate a hard negative prompt, stable diffusion3, and an image editing method (MagicBrush) to make pairs of positive and hard negative samples. Finally, they train a Stable diffusion model with an additional contrastive loss using the negative images on the denoising encoder representation. This leads to better compositional generation.

**Strengths:**

- Introducing a contrastive loss in the denoising encoder representation is an interesting idea.
- The experimental setups are clear, and there are good comparisons to baselines and ablations on their design choices.

**Weaknesses:**

- A bit more detail on how the contrastive loss is incorporated would be nice. Is the contrastive loss of the denoising encoder representation at any arbitrary time step t? I am assuming while training, the denosing encoder just predicts the noise to be subtracted for an arbitrary time step t. Hence, the losses would be the MSE on the noise prediction and the InfoNCE loss? What proportions are the added in?

- A few other datasets with hard-negative compositional images exist, albeit small. Two examples are COLA (https://arxiv.org/abs/2305.03689, they have a hard negative attribute-object composition image for a caption) and Winoground (which the authors already note).  A nice baseline to show would have been comparing using the author's synthetic dataset for tuning over using a combination of the real hard negative image datasets.

**Questions:**

See weaknesses.

---

> ### Author Response · Authors · 2024-11-20
> **Response to Reviewer drFx**
>
> We thank Reviewer drFx for their time and feedback! We are encouraged that R2 finds that “our contrastive loss is interesting” and “our experiment setups are clear”.  We will respond point-by-point to their comments below:
>
> >**W1: A bit more detail on how the contrastive loss is incorporated would be nice.**
>
> Thank you for this valuable feedback! Yes, we use latent encoder representation of positive and negative images at any arbitrary time step t, and apply a small projection head to map the image features to the space where the contrastive loss is applied. During training, the model predicts the noise to be subtracted. After experimenting with different proportions, we found that a weight of 0.1 for the InfoNCE loss yields the best performance.
>
> We believe it’s important to keep our framework clear and precise and will add more details in our revision.
>
>
> >**W2: Comparison with other datasets with hard-negative compositional images.**
>
> How our model would fare with other hard-negative dataset is indeed an interesting question.We include the results of fine-tuning our model with different datasets, evaluated by T2I-CompBench below..
>
> | Dataset | Color | Shape | Texture | Spatial | Non-Spatial | Complex |
> | ------- | ------ | ------ | ------- | ------- | ------- | ------- |
> | COLA | 62.20 | 48.98 |53.73 | 15.21 | 30.87 | 33.15 |
> | Winoground |  52.45 | 44.76 | 49.58 | 14.78 | 31.08 | 33.08 |
> | Ours | 71.04 |  54.57 | 72.34 | 21.76 |  33.08 |  42.52 |
>
> Although COLA attempts to construct **semantically** hard-negative queries, the majority of the retrieved image pairs are quite dissimilar in practice, often introducing a lot of noisy objects/background elements from the real images, due to the nature of retrieval from existing dataset. We hypothesize this makes it difficult for the model to focus on specific attributes/relationships in the context of compositionality. We show some examples in Figure 12. Furthermore, they don’t include complex prompts with multiple objects/attributes, and do not account for non-spatial relationships.
>
> Winoground contains high-quality contrastive images, however it only contains 400 effective sample pairs, covering 5 categories, which is too small for the model to learn and we believe the comparison is not fair.
>
> In contrast, our dataset ensures the generated image pairs are contrastive with **minimal visual changes**, enforcing the model to learn subtle differences in the pair, focusing on a certain category.
>
> We really appreciate your suggestion and agree comparing with other similar dataset is important and have added the above results in our revision (Appendix A.5)!

---

> ### Author Response · Authors · 2024-11-24
> **Follow-up to Rebuttal**
>
> Dear Reviewer drFx,
>
> Thank you again for your valuable feedback and acknowledgement of our work! We'd like to check in with you and see if our rebuttal has clarified your concerns. If there are still any unclear parts in our work, please let us know and we would be happy to engage further before the discussion period closes. Thanks for your time and effort in reviewing this paper!

---

> ### Comment · Reviewer_drFx · 2024-11-25
>
> Thank you for the review. The experiments on comparing to tuning with other kinds of hard negative data makes sense and highlights the befit of their data. I will keep my rating.

---

> > ### Author Response · Authors · 2024-11-26
> > **Response to Reviewer drFx**
> >
> > Thank you for recognizing our paper. We sincerely appreciate you reviewing our paper!

---

### Official Review · Reviewer_PM7r · 2024-11-05

**Soundness:** 3
**Presentation:** 3
**Contribution:** 3
**Rating:** 8
**Confidence:** 2

**Summary:**

This paper addresses a complex text-to-image generation approach that aims to achieve accurate and high quality generations of complex samples.  The authors develop a data generation pipeline where they use a LLM and a VQA model to create fine-grained differences in images to help train their mode.  The authors then use curriculum learning to train a diffusion model using their data.  The authors evaluate their approach using compositional T2I benchmarks and provide a user study of their approach.

**Strengths:**

1. The approach is well motivated and addresses a critical shortcoming of T2I models
2. The paper is relatively well written
3. The experiments cover a number of key comparisons, including a users study, to help validate the effectiveness of their approach

**Weaknesses:**

1. The data pipeline relies on VQA method's ability to measure alignment.  However, just as T2I models struggle with this, VLMs also struggle, e.g., [A,B].  As such, it seems quite questionable that this would provide a high quality dataset. The statement on L259 on quality control is also ambiguous. How many total annotators were used? How many samples?  How were the samples selected (i.e., how did you ensure coverage of all the elements of the dataset?). The sentence there is vague and gives no details.

2. The authors motivate the need to use generated data due to the cost of collecting pairs.  While I tend to agree with this generally, in practice it isn't clear if this is true.  Some benchmarks like [B] show that fine-grained negatives can be extracted from existing datasets.  There are also datasets with fine-grained pairs of images [C].  What would make this stronger is if the authors compared directly against trying to use some of the competing datasets in their model (including balancing the number of samples).  This would help demonstrate that the differences are important, or the effect of using generated vs. real pairs

3. The user study in figure 8 is not convincing that the proposed approach is better.  The alignment has improved, but at the cost of the generation quality.  I don't have a specific recommendation here- it seems to be simply a weakness of the approach.  That said- the discussion could be more forthcoming and perhaps discuss why despite the drop in aesthetics this is still important.

[A] See, Say, and Segment: Teaching LMMs to Overcome False Premises. CVPR 2024

[B] Cola: A Benchmark for Compositional Text-to-image Retrieval. NeurIPS 2023

[C] Evaluating Text-to-Image Matching using Binary Image Selection (BISON). 2019

**Questions:**

I proposed some specific questions in the weaknesses.  Generally I would mostly like to hear about W1 and W2

---

> ### Author Response · Authors · 2024-11-20
> **Response to Reviewer PM7r**
>
> We thank Reviewer PM7r for their time and feedback! We are encouraged that R1 finds that “our approach is well motivated” and “our experiments are comprehensive”.
> We will respond point-by-point to their comments below:
>
> >**W1:The data pipeline relies on VQA method's ability to measure alignment ...VLMs also struggle with this,...**
>
> We agree that the performance of our approach partially depends on the reliability of question-answering from the VQA model. Nevertheless, as shown through our quantitative results, significant improvements from prompt-image similarity (measured by CLIP score in Figure 4.) and compositionally ability evaluated by T2I-CompBench (Table 3) are obtained in our current framework by using VQA models.
>
> To analyze the accuracy of the VQA model's answering results, we sampled 1500 images (about 10% of the dataset, across 3 stages) and performed an additional user study wherein for each question-image pair, the human subject is asked to answer the same question. The accuracy of the VQA model is then predicted using the human labels as ground truths.
> Results are shown below (more details about data quality evaluation are in Appendix A.6).
>
> | Image Stage | VQA Accuracy % | Annotation Time / Image (s) |
> | ----- |---------------- | -------------------- |
> | Stage-I | 93.1% | 8.7s |
> | Stage-II | 91.4% | 15.3s |
> | Stage-III | 88.9% | 22.6s |
>
> We observe that the VQA model is able to effectively measure alignment even when the text prompt becomes complex. After filtering images with low alignment score, we use CLIPScore as a second check to select the best aligned image, further guaranteeing the quality of our dataset.
>
> - **L259 on quality control is also ambiguous**
>
> We tasked 3 annotators in total, each annotator assigned 2550 images out of the total 7650 images (about 50% samples, proportionally across three stages), to check if the positive and negative images align with the text prompt well and are similar with minimal changes. We filtered only 647 images from the selected 7650 images, which is 8.45%, attesting the quality of the dataset.
> The high-level objective of our paper is to propose an automatic pipeline to generate faithful contrastive image pairs, which we find crucial for guiding models to focus on compositional discrepancies. Especially with the advancement of VQA models and LLM models, our pipeline and dataset can benefit from higher alignment scores.
>
> As it is critical to keep our words precise, we have added clarification in our revision (Appendix A.6).
>
> >**W2: Comparison between synthesized vs. real hard negative dataset**
>
> How our model would fare with a real hard-negative dataset is indeed an interesting question.We include the results of fine-tuning our model with COLA, BISON evaluated by T2I-CompBench below (randomly sampled consistent number of samples across datasets).
>
> | Dataset | Color | Shape | Texture | Spatial | Non-Spatial | Complex |
> | ------- | ------ | ------ | ------- | ------- | ------- | ------- |
> | COLA | 62.20 | 48.98 |53.73 | 15.21 | 30.87 | 33.15 |
> | BISON | 59.49 | 49.36 |48.77 | 14.64 | 31.25 | 32.91 |
> | Ours | 71.04 |  54.57 | 72.34 | 21.76 | 33.08 | 42.52 |
>
> Although COLA and BISON attempt to construct **semantically** hard-negative queries, the majority of the retrieved image pairs are quite dissimilar in practice, often introducing a lot of noisy objects/background elements from the real images, due to the nature of retrieval from existing dataset. We hypothesize this makes it difficult for the model to focus on specific attributes/relationships in the context of compositionality. We show some examples in Figure 12. Furthermore, they don’t include complex prompts with multiple objects/attributes, and COLA does not account for non-spatial relationships.
> In contrast, our dataset ensures the generated image pairs are contrastive with **minimal visual changes**, enforcing the model to learn subtle differences in the pair, focusing on a certain category. To the best of our knowledge, no real contrastive image dataset only differs on minimal visual characteristics.
>
> We agree the comparison is very valuable and have incorporated the results in our revision (Appendix A.5)!
>
> >**W3: The user study in figure 8 is not convincing that…, but at the cost of the generation quality.**
>
> Since our model is fine-tuned on Stable Diffusion models, which tends to generate lower aesthetic scores compared to DALLE-2 and PixArt-Alpha [1], it is important to note that our method does not compromise aesthetic quality in comparison to SD v3 and SDXL–as shown in Figure 8, after fine-tuning, our method outperforms both in terms of aesthetic quality and alignment.
>
> [1] Chen, et.al., PixArt-α: Fast Training of Diffusion Transformer for Photorealistic Text-to-Image Synthesis

---

> ### Author Response · Authors · 2024-11-24
> **Follow-up to Rebuttal**
>
> Dear Reviewer PM7r,
>
> Thank you again for your valuable feedback and acknowledgement of our work. We'd like to check in with you and see if our rebuttal has clarified your concerns. If there are still any unclear parts in our work, please let us know and we would be happy to engage further before the discussion period closes. Thanks for your time and effort in reviewing this paper!

---

> ### Comment · Area_Chair_xL2s · 2024-11-28
> **Reviewer feedback and discussion**
>
> Dear Reviewer PM7r,
>
> As the discussion period will close next week, please take some time to read the authors' rebuttal and provide feedback as soon as possible. Did the author address your concerns, and do you have further questions?
>
> Thanks,
>
> Area Chair

---

> ### Comment · Reviewer_PM7r · 2024-11-28
>
> Thanks, I don't have any other questions at this time.  I increased my score to a 8, thanks for the new experiments and discussion.

---

> > ### Author Response · Authors · 2024-11-29
> > **Response to Reviewer PM7r**
> >
> > We sincerely appreciate your thorough review of our work and your acknowledgment of the efforts we have made to address your concerns!

---

### Author Response · Authors · 2024-11-21
**General Repsonse**

We thank all the reviewers for their encouraging evaluations and constructive feedback. We underscore five key contributions of our paper mentioned by the reviewers:

1. Our paper addresses a critical issue in T2I models, focusing on the challenge of compositionality (**PM7r, cqxf, Amfr**).
2. Our approach is well-motivated, novel and our paper is well-written  (**PM7r, Amfr**).
3. We introduce an effective fine-tuning strategy with contrastive loss and multi-stage training on diffusion models and empirically demonstrate its improved performance (**drFx, cqxf**).
4. We construct a valuable new dataset that features contrastive image pairs with only specific changes for future research (**cqxf, Amfr**).
5. Our experiments are comprehensive and clear (**PM7r, drFx, cqxf**).

**We have updated the revision PDF and highlighted the changes in our appendix. Here, we summarize our changes in the manuscript:**

* **User studies to evaluate the dataset quality**

Following the suggestion of Reviewer PM7r and Amfr, we conduct additional user studies to evaluate the effectiveness of our dataset construction framework. We added the details about human evaluation including interfaces in Appendix A.6.  The evaluation contains the following three parts:
1. Accuracy of question-answering of VQA models
2. Coverage of LLM-generated QA pairs
3. Alignment of revised caption with images

- **Remarks of dataset differences**

Motivated by Reviewer PM7r and drFx, we included the comparison or our COM-DIFF dataset with other hard-negative image dataset in Appendix A.5.


- **Illustration of dataset construction procedure**

As suggested by Reviewer cqxf, we have included a discussion on the text prompt generation procedure, particularly emphasizing how we avoided overlaps with other benchmarks, in Appendix A.1.

To empirically show how captions are revised to align images during data construction (Amfr), we elaborate the instructions and show examples in Appendix A.3.


- **Evaluation on a different benchmark**

In Appendix C.2, we have added further comparisons with a recent T2I compositionality benchmark, as recommended by Reviewer cqxf.

- **Related work**

In response to Reviewer cqxf, we incorporated a subsection on relevant research to provide a richer context in Appendix E.

We hope this addresses the main concerns raised by the reviewers, and we welcome any additional feedback or new concerns for discussion!

---

### Meta-Review · Area_Chair_xL2s · 2024-12-19

**Metareview:**

This paper presents a novel approach for compositional text-to-image generation by contrastive images. This paper receives consistent positive reviews from all of the four reviewers, recognizing the strengths of strong motivation, method design, and new dataset. Questions are raised on datasets, benchmark, and experiments. The authors addressed the questions in the rebuttal.

**Additional Comments On Reviewer Discussion:**

The authors addressed the questions on datasets, benchmark, and experiments during the rebuttal. All of the reviewers responded and recognize the authors' efforts in the rebuttal.

---

### Decision · Program_Chairs · 2025-01-22

Accept (Spotlight)